

# A sanity check for earthquake recurrence models used in PSHA of slow deforming regions: the case of SW Iberia

Margarida Ramalho[1], Luis Matias[2], Marta Neres[3], Michele M. C. Carafa[4], Alexandra Carvalho[5], and Paula Teves-Costa[2]

[1]Faculdade de Ciências da Universidade de Lisboa, Campo Grande, 1749-016 Lisbon, Portugal

[2]Instituto Dom Luiz (IDL), Faculdade de Ciências da Universidade de Lisboa, Campo Grande, 1749-016 Lisbon, Portugal

[3]Instituto Português do Mar e da Atmosfera, Lisbon, Portugal

[4]Istituto Nazionale di Geofisica e Vulcanologia, Sezione di Tettonofisica e Sismologia, L'Aquila, Italy

[5]LNEC, Avenida do Brasil 101, 1700-066 Lisbon, Portugal

*Correspondence to:* M. Ramalho (margarida_ramalho@hotmail.com)

**Abstract.** Probabilistic Seismic Hazard Assessment (PSHA) is the most common tool used to decide on the acceptable seismic risk and corresponding mitigation measures. We propose two consistency tests to address the variability of earthquake generation models found in PSHA studies: i) one rule-of-thumb test where the seismic moment release from the model is converted to an average slip on a typical fault and compared with known plate kinematics or GNSS deformation field; ii) using 15 a neotectonic model, the computed deformation is converted into seismic moment release and to a synthetic earthquake catalogue. We apply these tests to the W and SW Iberia slow deforming region, where two earthquake source areas are investigated: 1) the Lower Tagus Valley, one of the largest seismic risk zones of Portugal; and 2) the offshore SW Iberia area, considered to be the source for the 1st November 1755 event (M~8.7). Results show that some of the earthquake source models should be considered as suspicious, given their high/low moment release when compared to the expected values from GNSS 20 observations or neotectonic modelling. In conclusion, PSHA studies in slow deforming regions should include a similar sanity check on their models' evaluation, downgrading the weight of poorly compliant models.



## 1. Introduction

In earthquake-prone areas modern societies with limited available resources must decide on the level of acceptable risk to define the appropriate mitigation measures. Probabilistic Seismic Hazard Assessment (PSHA) is the most common tool used by decision-makers to attain this equilibrium between risk and protection. Many PSHA studies have been conducted for emergency planning, land use management or to help establishing or revising building codes. Engineers use PSHA to help defining the ground motion levels in building codes that will avoid failure or collapse of constructions in the case of some

expected earthquake occurs. PSHA was established by Cornell (1968) and since then has seen a large evolution and wide-spread application. PSHA is a technical methodology in seismology, but its use has political and economic constraints that must be addressed by society as a whole.

PSHA, in its simpler description, provides the probability that a given level of ground motion, $U_0$, may be exceeded at a given location in a specific time period T, usually taken as the life-time of the structures concerned. For a single site with a single

earthquake source at a fixed distance, the result of PSHA is the product of two probabilities (e.g. Frankel, 2004),

$$P(U \geq U_0|T) = P(E|T) \times P(U \geq U_0|E), \tag{1}$$

Here $P(E|T)$ is the probability that the earthquake occurs during the time period T, and $P(U \geq U_0|E)$ is the probability of ground shaking exceeding the reference value $U_0$ if the earthquake occurs. This simple description is useful since it splits the PSHA work in two different problems. The first factor is controlled by our knowledge of earthquake generation, while the

second factor is addressed by the choice of Ground Motion Prediction Equations (GMPEs) and its associated uncertainties. One full PSHA study must consider all the possible earthquake sources affecting each site and Eq. (1) becomes an integral of all possible contributions.

One of the evolutions suffered by PSHA studies and now recognized as essential, is the evaluation of the uncertainties on the results (e.g. Frankel, 2004, Stein et al., 2012, Mulargia et al., 2017), being a consequence of our incomplete knowledge of the

earthquake generation and propagation mechanisms. They can be classified as aleatory, the uncertainty in random effects, and epistemic, the uncertainty in knowledge (uncertainty about the best model to use in PSHA computations and several models are possible). Uncertainties are already included in standard PSHA where GMPEs are defined as statistical laws and the standard deviation of the logarithm of the predicted parameter is usually computed. Aleatory uncertainties in other parameters can be included by Monte-Carlo simulations and epistemic uncertainty is depicted through a logic tree, where each alternative

model is ascribed some weight for PSHA computation. It is important to say that the weighting factors are usually assigned subjectively, often using expert opinion.

As pointed out by several authors (e.g. Stein et al., 2012, Mulargia et al., 2017) recent large earthquakes have occurred in areas where the seismic hazard was considered low or they caused ground shaking much greater than the one expected by PSHA studies. These events have demonstrated the weaknesses of PSHA and a deficit in the evaluation of its uncertainties. As shown

by Stein et al. (2012) there are several lines of research to help on improving the reliability of PSHA studies.





The results of the PSHA studies are expressed as a statistical parameter on the exceedance probability for the ground motion parameter chosen. Results from different studies can only be compared if the final parameters to express PSHA are the same (e.g. peak ground motion with 10% probability of being exceed in 50 years). When comparable, the differences among different studies' results can be huge. This brings confusion and uncertainty to the users of these studies, who use results for emergency

management or land use planning and question the foundations for establishing or revising building codes. This large diversity is usually ascribed to the large uncertainty on the 2$^{nd}$ term of Eq. (1), the uncertainty and differences in the Ground Motion Prediction Equations (GMPEs) used in the studies. However, there is also a large diversity on the earthquake generation models (first term of Eq. (1)) and on the ways that the related uncertainties are assessed, but its consequences on the final PSHA results are difficult to assess and usually overlooked.

In this paper, we address the first factor of Eq. (1), the earthquake generation process, and propose a methodology to evaluate and validate earthquake generation models that are used for PSHA studies. This methodology is applied to the western Iberia (Fig. 1), which is the source area of the largest earthquakes felt in Europe since year 1000. This area can be considered a typical example of Slow Deforming Regions (SDR) where the instrumental and historical catalogues are incomplete when compared to the large (and uncertain) return periods of the large earthquakes, and where active faults are challenging to identify since

erosional processes exceed the tectonic activity. Furthermore, reverse faults may develop as splays or blind thrusts that rarely reach the surface. Therefore, SDR are the most challenging domains to make PSHA studies and where they are more likely to fail. However, if it is possible to measure or estimate the rate of strain accumulation, then the earthquake generation models used in PSHA that exceed that rate or that are far below, should be considered as suspicious and its likelihood degraded accordingly.


**Figure 1:** Geodynamic setting of the investigated area. a) Seismicity in and around Europe. Yellow dots show earthquake epicentres with M≥5.0 from ISC (2018) for the period 1900-2018. Red stars show epicentres of events with M≥8.0 according to the European instrumental






(Grünthal and Wahlström, 2012) and historical catalogues (Stucchi et al., 2013). b) The plate boundary setting of the studied area. The light-yellow domain indicates a diffuse plate boundary. c) Active and probable active faults identified in Western Iberia mainland and offshore (modified from Cabral, 2012 and Duarte et al., 2013).

Despite the recent failures and criticisms on PSHA and its foundations, we consider that the earthquake generation models used in PSHA studies can be evaluated using the elastic rebound principles, i.e., earthquakes are generated by accumulated strain in the lithosphere.

In more active areas, like subduction zones, a more complex interplay between seismicity and strain is possible. Carafa et al. (2018) investigating the Calabrian Arc proposed that the low strain rates observed by GPS together with large seismic moment release may be explained by a model with high interseismic coupling and low seismic coupling of the subduction interface. Meaning that some of the regional seismic strain is released by slow earthquakes that could induce clustered normal fault events on the upper plate.

Strain rates can result from tectonic stresses due to plate motions (near or far from plate boundaries), can result from glacial isostatic adjustment (GIA, Peltier et al., 2015) or may be due to gravity potential energy (GPE) differences. GPE contribution to the state of stress in Iberia was evaluated by Neres et al. (2018) and they conclude that GPE is a second order effect that is mostly identified in high relief areas, like the Pyrenees and Cantabria mountain ranges. In this work, we consider that in the investigated domains of W and SW Iberia, earthquakes are generated only by tectonic stresses controlled by the horizontal strain rate. If the strain rate is estimated, it can be used as a ruler to compare with the earthquake generation models that are proposed in PSHA studies, thus providing a geodynamic constraint. We argue that the proposed methodology should be implemented in PSHA studies as a sanity test for competing earthquake generation models, in particular for other areas of the globe characterized by slow and/or diffuse deformation.

## 2. Geodynamic setting

The tectonic activity affecting the Western and Southwestern Iberia is mainly caused by the slow convergence between Eurasia and Africa (Nubia) plates at a rate ~4 mm/year (e.g. Fernandes et al., 2003, Fig. 1a; Neres et al., 2016). The relative plate movement is accommodated by several tectonic features in what can be considered as a diffuse plate boundary (e.g. Duarte et al., 2013, Neres et al., 2016, Fig. 1c). Despite the slow convergence rate, the SW Iberia has been the locus of the largest magnitude earthquakes that destroyed Europe, like the 28th February 1969 (Mw=8.0, Grünthal and Wahlström, 2012) and the 1st November 1755 Lisbon earthquake with magnitude of 8.7 (Stucchi et al., 2013, Fig. 1a). The tectonic complexity of the area is compounded by another stress engine that has been invoked to affect SW Iberia, namely the westward movement of the Alboran domain, driven by subduction roll-back in the Gulf of Cadiz (e.g. Neres et al., 2016 and references therein). The



tectonic stresses generated by the plate convergence and slab roll-back propagate inland generating also some significant
earthquakes, like the 23rd April 1909 M~6.1 (e.g. Teves-Costa et al., 2019).

The Western and Southwestern Iberia are usually considered as one Stable Continental Region (SCR) onshore domain
(Johnston, 1989) in the sense that the most recent active tectonic event, the Alpine orogeny that led to the formation of the
Pyrenees and the Betics in Iberia, attained its activity peak in Eocene and Miocene times (e.g. Terrinha et al., 2019). In W and
SW Iberia, the Alpine orogeny resulted in the inversion of meso-cenozoic basins in central Portugal (the Lusitanian basin) and
in southern Portugal (the Algarve basin) with a geographically limited expression (ibid.). The current tectonic activity observed
inland is thus typical of Slow Deforming Regions (Custódio et al., 2015) with fault slip rates smaller than 1 mm/year.

Geologists consider as active faults the structures that show evidence of recent displacement (geological, geophysical,
historical) so that new displacements may be expected in the future. In W and SW Iberia, the slip rate measurements in the
identified active faults are small, from 0.005 mm/year to a maximum of 0.2 mm/year (Cabral, 1995, 2012). This implies that
the classification of tectonic features as active is very conservative. Any structure showing activity since the upper Pliocene
or Quaternary is considered as active (roughly since the last 3 Ma, Fig. 1c). In the offshore domain, closer to the plate boundary
between Eurasia and Africa (Nubia) the identified active or probable faults (Duarte et al., 2013, Fig. 1c) display evidence of
more recent activity, probably since the Holocene (last 10 ky).

The last decade has seen a considerable improvement in the seismic monitoring network allowing for a very detailed location
of earthquakes to magnitudes down to 1 (Custódio et al., 2015). The seismicity pattern that emerged from the more recent and
best-located events shows that most of the earthquakes occur in clusters and seismic belts roughly oriented NNE-SSW and
WNW-ESE. However, the relationship between these events and geologically mapped faults remain elusive (ibid.) in most
cases, even after a 3D relocation using a regional tomographic model (Veludo et al., 2017). In offshore SW Iberia, the source
area for the large 1755 and 1969 magnitude earthquakes (Fig. 1a), seismicity relocated by Ocean Bottom Seismometers (OBS)
has shown that most of the small magnitude events are occurring at mantle depths, down to 50 km focal depths, again without
a clear relationship with the known active tectonic features identified by geological and geophysical surveys (Geissler et al.,
2010, Grevemeyer et al., 2016, Silva et al., 2017).

Despite the high detail of the tectonic map in the Gulf of Cadiz, SW Iberia, that resulted from the extensive investigation by
geological and geophysical surveys (e.g. Duarte et al., 2013, Fig. 1c) the origin of the 1st November 1755 destructive earthquake
and tsunami remains uncertain with several proposals made by different authors (see Ribeiro et al., 2006, for a revision). The
other large earthquake originated in the region, the 28th February 1969 (Mw=8.0), despite the well-constrained epicentre, focal
depth and source mechanism (Fukao, 1973), could not be ascribed to any active fault identified in the area (Martínez-Loriente
et al., 2013).





**3. Earthquake generation models used in PSHA studies for W and SW Iberia**

Probabilistic Seismic Hazard Assessment studies have been used in the last decades in W and SW Iberia as a tool for emergency planning, land use management and as an input to be considered in the elaboration of building codes (e.g. Vilanova and Fonseca, 2007, Campos Costa et al., 2008, Mezcua et al., 2011, IGN-UPM, 2013, Woessner et al., 2015). Some of these studies had a global scope, like the GSHAP (Giardini, 1999) or the 2013 European Seismic Hazard Model (ESHM13, Woessner et al., 2015, Fig. 2a), others were conducted to support the elaboration or revision of national building codes (Campos Costa et al., 2008, IGN-UPM, 2013, Fig. 2b). For land use management and emergency planning, more local and regional studies have been conducted (e.g. for Portugal, Carrilho et al., 2010, Matias and Teves-Costa, 2011a, b, for SE Spain, Rivas-Medina et al., 2018). All the above mentioned PSHA studies use different earthquake generation models with a common feature, time independence. In active seismic zones, such as those associated with subduction, or in areas where there is almost complete knowledge of the active structures, it is possible to carry out PSHA studies in which the earthquake recurrence is a function of time. This is not true for the slow deforming region of W and SW Iberia.

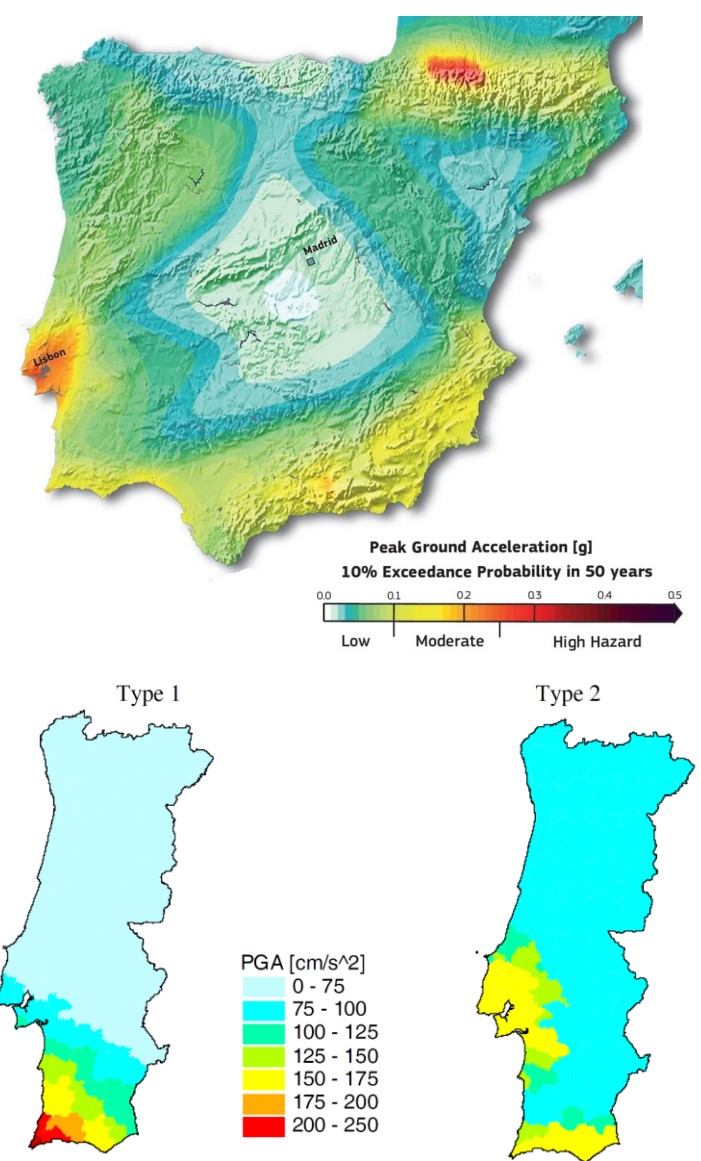

**Figure 2:** Seismic hazard maps expressed as Peak Ground Acceleration (PGA) expected to be exceeded in 50 years with 10% probability. (a) From the ESHM13 model by Woessner et al. (2015), units in g; (b) From the PSHA study used in Portugal for the revision of the building codes according to EC8 guidelines (Campos Costa et al., 2008), units in cm/s$^2$. Type 1 refers to a "far" (offshore) source while type 2 refers to a "near" source.

Another consequence of the geodynamic setting of the study area is that the preferred earthquake source model is the zone source where earthquakes are assumed to occur with a uniform probability anywhere inside the area considered. This approach is justified for the onshore where the known fault slip rates is very low (see section 2). On the offshore the plate boundary is



complex without a clear relationship between current seismicity and the tectonic faults identified by geological and geophysical campaigns (e.g. Zitellini et al., 2009; Custódio et al., 2015; Neres et al., 2016), as detailed in section 2. Recently, in smaller domains where the fault geometry and activity are considered to be well known, like in SE Spain, a mixed source model has been proposed (Rivas-Medina et al., 2018) where the earthquake potential is distributed between the faults and the zone. This

is an exceptional case that cannot be extended to the whole W and SW Iberia.

The choice of the source generation zones for PSHA is done using geological criteria, distribution of active faults, nature and thickness of the crust, and geophysical criteria, such as earthquake epicentre distribution, location and macroseismic fields of historical earthquakes. However, there is no common agreement on the geological and geophysical information to be used, nor on the weight that is applied to each of the criteria, resulting in a wide diversity of source zones for the W and SW Iberia (Fig.

175   3).



**Figure 3:** Seismogenic zones proposed for the 5 PSHA studies analysed in this work. The zones corresponding to Lower Tagus Valley (LTV) sources and 1755 earthquake selected for sanity check are outlined in black. a) the National Annexes of Eurocode 8 (EC8), adapted from Campos Costa et al. (2008). b) the ERSTA study, adapted from Carrilho et al. (2010). c) the SHARE study, adapted from Woessner et

al. (2015). d) the QREN study, adapted from Matias and Teves-Costa (2011a, b). e) and f) adapted from Vilanova and Fonseca (2007) study. e) is seismic zonation SA and f) is seismic zonation SB.

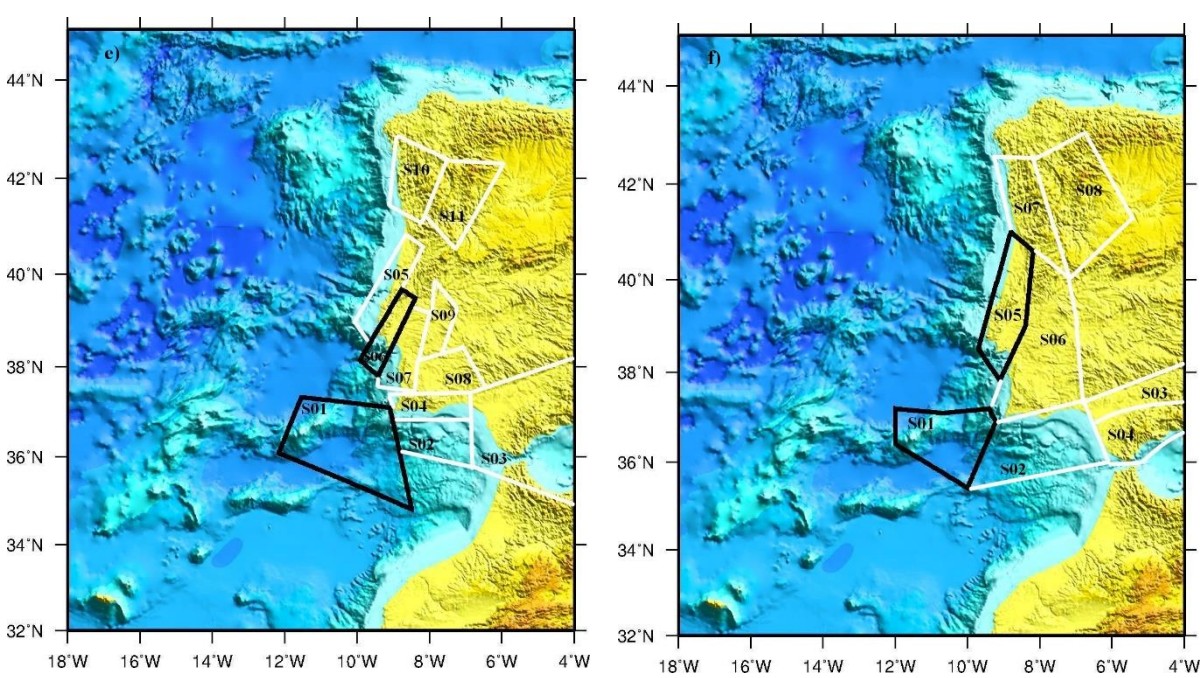

**Figure 3 (continued):** Seismogenic zones proposed for the 5 PSHA studies analysed in this work.


On each source zone, the earthquake recurrence is expressed by the double truncated Gutenberg-Richter law (DTGR)

$$\dot{N}(m) = \lambda \frac{e^{-\beta(m-m_{min})} - e^{-\beta(m_{max}-m_{min})}}{1 - e^{-\beta(m_{max}-m_{min})}} \tag{2}$$

Where,

$\dot{N}(m) = \lambda_m$      Number of earthquakes with magnitude $\geq m$ per year

$m_{min}$      Minimum magnitude to be considered

$\lambda$      Number of earthquakes per year with magnitude $\geq m_{min}$

a, b      Usual parameters for the Gutenberg-Richter law, $\log\left(\dot{N}(m)\right) = a - bm$

         a can be referred to $m = m_{min}$ or to $m=0$. On the first case we have

         $\lambda = 10^{a-bm_{min}}$ On the second case we have $\lambda = 10^a = e^\alpha$, with

195           $\alpha = a \ln(10)$

$m_{max}$      Maximum magnitude considered for the source zone





β    Asymptotic slope for the DTGR law. β=*b ln10*

The definition of earthquake recurrence on each source zone requires the definition of the set of parameters $(m_{min}, m_{max}, \alpha, \beta)$ or $(m_{min}, m_{max}, a, b)$. These parameters are estimated by criteria defined on each PSHA study and may vary considerably

200 depending on the choice of the earthquake catalogues to use (instrumental and historical, declustered or not and declustering methods), the methods used to obtain the $(a, b)$ from an incomplete seismic catalogue (least squares, maximum likelihood, etc.,) and the approach used to define the maximum magnitude to be considered.

Some of the PSHA studies mentioned above do not consider any uncertainty in the definition of the earthquake generation model (zoning and earthquake recurrence), others consider that different models can be accepted with some weights, with

205 uncertainty assessed by the logic tree method (e.g. Vilanova and Fonseca, 2007, Woessner et al., 2015), while others also consider that the DTGR parameters have a statistical uncertainty that is assessed by Monte-Carlo simulations (e.g. IGN-UPM, 2013, Gaspar-Escribano et al., 2015).

In this paper we will not discuss or compare the methodologies used in defining the earthquake generation models that have been published for W and SW Iberia. Instead, we will use the published parameters to make a sanity check by comparing the

210 seismic moment release that results from them with the presumed tectonic deformation. For this investigation we consider 5 different PSHA studies: i) the study used in Portugal for the revision of the building codes according to EC8 guidelines (Campos Costa et al., 2008; hereafter referred as EC8); ii) the regional study by Carrilho et al. (2010) used for emergency management in Southern Portugal (hereafter referred as ERSTA); iii) the European scale ESHM13 model by Woessner et al. (2015) (hereafter referred as SHARE); iv) the local study by Matias and Teves-Costa (2011a,b) used for emergency

215 management in Lisbon and Cascais (hereafter referred as QREN), and; v) the Vilanova and Fonseca (2007) study used for the evaluation of seismic risk in Portugal mainland by Silva et al. (2015) (hereafter referred as VF2007). We will concentrate our efforts only on two earthquake generation zones: the one that affects Lisbon, for the high risk implied and because it is the area where large differences in PSHA are evident (Fig. 2); and the offshore domain in the Gulf of Cadiz that is considered by different studies as the source area for the M~8.7 1ˢᵗ November 1755 earthquake.


### 3.1 The EC8 model

In 2006, for the preparation of the National Annex of Eurocode 8 (project regulation for the construction of structures for earthquake resistance), the seismic hazard was assessed considering eleven seismogenic zones for Portugal Mainland (Fig. 3a) and the PSHA results for Portugal Mainland, Azores and Madeira Archipelagos are presented in Campos Costa et al. (2008).

225 For Portugal Mainland the main criteria used for the definition of seismic source areas was the distribution of historical and instrumental seismicity, with seismotectonic considerations as secondary criteria.



The details of the seismogenic zoning can be found in Sousa (1996) but in general, the outlined zones fall into two wide categories: interplate zones (zones 6, 7 and 9 of Fig. 3a), which originate earthquakes with an epicentre mainly located on the plate, and the intraplate zones, which originate earthquakes with an epicentre located predominantly inside the Eurasian plate
(zones 1, 2, 3, 4, 5, 8, 10 and 11 of Fig. 3a). The parameters that characterize these zones were obtained after studying the exhaustiveness of the catalogue (where the minimum magnitude or exhaustiveness magnitude is defined), considering earthquakes with a magnitude equal to or greater than 3.5.

### 3.2 The ERSTA model

For the project ERSTA, "*Estudo do Risco sísmico e de Tsunamis do Algarve*", financed by the Portuguese Civil Protection Authority (ANEPC), a concerted work was carried out by several national entities that, after reviewing the seismic, tectonic and geological information of the Algarve region, allowed to redesign the seismogenic zones (15 zones in total) that potentially affect the region (Carrilho et al., 2010). These zones are shown on Fig. 3b. With the idea of considering seismic hazard for the entire continental territory, and in the scope of other local studies (Jeremias et al., 2012), 4 more seismogenic zones were
defined, modelled by large areas due to the difficulty in connecting the epicentres to the mapped faults in the neotectonic map.

An exhaustiveness study was carried out (Rodrigues et al., 2009) for the new seismic catalogue, reviewed by the IPMA (*Instituto Português do Mar e da Atmosfera*), and the parameters that characterize these areas were determined (Rodrigues et al., 2009; Jeremias et al., 2012; Carvalho and Campos Costa, 2015). In this model of seismic sources, area B (Fig. 3b) is part of the Cenozoic Basin of the Lower Tagus Valley, and zone J comprises the Horseshoe Fault, the Marquês de Pombal Fault,
and the São Vicente Fault. In the area J some of the most important earthquakes were generated, such as the one on February 28, 1969, and, most likely, the one on November 1, 1755.

### 3.3 The SHARE models

In 2013, the European Seismic Hazard Model (ESHM13), resulted from a probabilistic hazard assessment supported by the
EU-FP7 project SHARE (2009-2013). The ESHM13 is a consistent seismic hazard model for Europe and Turkey that overcomes the limitation of national borders and includes a complete quantification of uncertainties. It is the first complete contribution to the initiative called "Global Earthquake Model". And it can serve as a reference model for various applications, from preparing for an earthquake to strategies to mitigate seismic risk (Woessner et al., 2015) and, according to Carvalho and Malfeito (2016), to serve as the basis to the review of the National Annexes of Eurocode 8. Figure 3c shows the source areas
defined for W and SW Iberia by the SHARE project.

The earthquake recurrence laws used by SHARE, namely the a and b parameters of the Gutenberg-Richter law and the maximum magnitude for each zone, were proposed with an integrated and homogenization perspective of the model across





Europe. In this project, the epistemic uncertainties of the model components and the hazard results were accounted using a Logic Tree (Woessner et al., 2015). On each zone three values for the maximum magnitudes were considered, minimum, mean

and maximum, that we incorporate in our sanity check evaluation as SHAREmin, SHAREmed, and SHAREmax, respectively.

### 3.4 - QREN model

The seismic zonation referred to this model was established in the study carried out for the Assessment of Natural and Technological Risks in the City of Lisbon, named as QREN (*Quadro de Referência Estratégico Nacional*). In that work (Matias

and Teves-Costa, 2011a,b), the determination of the earthquake recurrence parameters was obtained considering the working earthquake catalogue declustered decomposed into one historical catalogue and 3 instrumental catalogues, selected according to the completeness magnitude estimated. The definition of the source areas took into consideration both the seismic activity and the distribution of the main active tectonic structures identified in the mainland territory of Portugal (Teves-Costa et al., 2001). Thus, it was proposed to distinguish 8 seismogenic zones (Matias and Teves-Costa, 2011a, b) shown on Fig. 3d.

In this model, the earthquake recurrence model was obtained using the HA2 application developed in MATLAB by Andrezej Kijko (Kijko and Sellevoll, 1992; Kijko, 2004). The HA2 combines information from several catalogues, using a Bayesian model, applying a Poisson-Gamma distribution for the occurrence of earthquakes in time and an Exponential-Gamma distribution for the magnitude distribution. The determination of $m_{max}$ magnitude is obtained by an iterative process, with the parameters $\lambda$ and $\beta$ obtained by maximum likelihood. The HA2 algorithm results in a probability distribution that does not

follow exactly the DTGR and so, to make it comparable with the others, a fit to DTGR had to be made.

### 3.5 The Vilanova and Fonseca (2007) models

Vilanova and Fonseca (2007) presented two proposals for dividing the territory into seismogenic zones: SA and SB (Fig. 3e, 3f). SA encompasses eleven seismogenic zones chosen based on seismicity criteria: in regions with moderate to large local

historical earthquakes, the boundaries of the seismogenic areas follow the VII (MMI scale) isoseismal considering that the area delimited by this relatively higher damage distribution includes most probably its geological source. In all other regions the source zones encompass areas of diffuse seismicity. The SB proposal consists in eight seismogenic zones adapted from Peláez and López Casado (2002).

Vilanova and Fonseca (2007) used a logic tree scheme to deal with epistemic uncertainties on ground motion prediction

equations, source areas, the catalogue used, the type or earthquake recurrence and the maximum magnitude. For the sanity check we deal only with the parameters that influence the earthquake recurrence, which are the ones presented in Fig. 4. Considering all possible branches, we will test 32 zoning and earthquake recurrence combinations. Each branch will be




identified by its components, SA or SB, RA or RB, a1 (M>4.0) or a2 (M>4.6), max0 or max+. When referred collectively, these models will be identified as VF2007.


**Figure 4:** Logic tree used for hazard calculation. The weight of each branch is shown in straight parentheses (modified from Vilanova and Fonseca, 2007).

**3.6 – Synthesis of earthquake recurrence models to be tested**

For the five analysed PSHA studies we selected from each one the source zones that correspond to the LTV and 1755 areas to be investigated (outlined in black in Fig. 3). The respective recurrence model parameters are summarized in Tables 1 to 4.






**Table 1:** Earthquake recurrence parameters for the Lower Tagus Valley (LTV) source zone from 4 of the models investigated.

| | Lower Tagus Valley (LTV) | | |
|---|---|---|---|
| Model | a | b | $m_{max}$ |
| EC8 | 2.41 | 0.71 | 7.2 |
| ERSTA | 3.03 | 0.79 | 7.1 |
| SHAREmin | 3.4 | 0.9 | 7.1 |
| SHAREmed | 3.4 | 0.9 | 7.4 |
| SHAREmax | 3.4 | 0.9 | 7.6 |
| QREN* | 2.46 | 0.88 | 6.87 |

* Best fit to DTGR law








**Table 2:** Earthquake recurrence parameters for the Lower Tagus Valley (LTV) source zone for the 16 combinations in the logic tree proposed by Vilanova and Fonseca (2007). Each combination has, in addition, two possible maximum magnitudes, resulting in 32 possible recurrence models.

| | | Lower Tagus Valley (LTV) | | | |
|---|---|---|---|---|---|
| Model | Options | $a$ | $b$ | $m_{max}$ | $m_{max}+0.5$ |
| Vilanova SA | SA-CA-RA-a1 | 2.88 | 0.94 | 6.9 | 7.4 |
| | SA-CA-RA-a2 | 3.03 | | | |
| | SA-CB-RA-a1 | 2.74 | 0.89 | | |
| | SA-CB-RA-a2 | 2.92 | | | |
| | SA-CA-RB-a1 | 3.31 | 1.02 | | |
| | SA-CA-RB-a2 | 3.42 | | | |
| | SA-CB-RB-a1 | 3.24 | 0.99 | | |
| | SA-CB-RB-a2 | 3.40 | | | |
| Vilanova SB | SB-CA-RA-a1 | 2.79 | 0.91 | 7.1 | 7.6 |
| | SB-CA-RA-a2 | 2.57 | | | |
| | SB-CB-RA-a1 | 2.79 | 0.91 | | |
| | SB-CB-RA-a2 | 2.57 | | | |
| | SB-CA-RB-a1 | 3.53 | 1.00 | | |
| | SB-CA-RB-a2 | 3.67 | | | |
| | SB-CB-RB-a1 | 3.58 | 1.02 | | |
| | SB-CB-RB-a2 | 3.70 | | | |

SA/SB – relative to the zonation used; CA/CB – relative to the catalogue used; RA/RB – recurrence model used; the a1 values were calculated
for m≥4.0 and the a2 for m≥4.6; considering two values of maximum magnitude.





**Table 3:** Earthquake recurrence parameters for the 1755 source zone from 4 of the models investigated.

| | 1755 Source zone | | |
|---|---|---|---|
| Model | a | b | $m_{max}$ |
| EC8 | 2.70 | 0.72 | 8.8 |
| ERSTA | 2.44 | 0.62 | 8.7 |
| SHAREmin | 4.0 | 0.9 | 8.5 |
| SHAREmed | 4.0 | 0.9 | 8.7 |
| SHAREmax | 4.0 | 0.9 | 8.8 |
| QREN* | 3.13 | 0.92 | 8.9 |

\* Best fit to DTGR law






**Table 4:** Earthquake recurrence parameters for the 1755 source zone for the 16 combinations in the logic tree proposed by Vilanova and Fonseca (2007). Each combination has, in addition, two possible maximum magnitudes, resulting in 32 possible recurrence models.

| | | 1755 Source zone | | | |
|---|---|---|---|---|---|
| Model | Options | $a$ | b | $m_{max}$ | $m_{max}+0.5$ |
| Vilanova SA | SA-CA-RA-a1 | 4.14 | 1.03 | 8.5 | 9.0 |
| | SA-CA-RA-a2 | 4.12 | | | |
| | SA-CB-RA-a1 | 3.77 | 0.94 | | |
| | SA-CB-RA-a2 | 3.72 | | | |
| | SA-CA-RB-a1 | 4.55 | 1.13 | | |
| | SA-CA-RB-a2 | 4.63 | | | |
| | SA-CB-RB-a1 | 4.46 | 1.10 | | |
| | SA-CB-RB-a2 | 4.53 | | | |
| Vilanova SB | SB-CA-RA-a1 | 3.78 | 0.94 | 8.5 | 9.0 |
| | SB-CA-RA-a2 | 3.72 | | | |
| | SB-CB-RA-a1 | 3.77 | 0.94 | | |
| | SB-CB-RA-a2 | 3.70 | | | |
| | SB-CA-RB-a1 | 4.35 | 1.09 | | |
| | SB-CA-RB-a2 | 4.34 | | | |
| | SB-CB-RB-a1 | 4.45 | 1.12 | | |
| | SB-CB-RB-a2 | 4.45 | | | |

SA/SB – relative to the zonation used; CA/CB – relative to the catalogue used; RA/RB – recurrence model used; the a1 values were calculated for m≥4.0 and the a2 for m≥4.6; considering two values of maximum magnitude.


## 4. Methodologies to evaluate the consistency between tectonic and seismic deformation

There are two main ways of modeling lithospheric deformation: the kinematic approach and the dynamic approach. In kinematic modeling long-term horizontal velocity field is constrained by simultaneously inverting all available geophysical information, such as GPS-derived velocities and stress data records. Models of this type rarely explain the fundamental tectonic mechanisms and leave no independent data sets available for postprocessing testing. Because it emphasizes the role of geodetic



data, it is often the preferred method for modeling on-land deformation (Bird, 2009; Carafa et al., 2020). However, it does not allow consistent estimation of the offshore long-term deformation, where the subjective choice of the active faults to insert into the model strongly determines the localization of deformation, with negligible influence left to other geophysical data. On the other hand, in the dynamic modeling approach the stress equilibrium equation is solved using estimated rock rheologies,

layer thicknesses, and boundary conditions, i.e., the velocity field is calculated from the known structure and physics of the Earth. In this case, several data sets, e.g. geodetic velocities, fault slip rates, seismic activity, and stress directions, can be used to assess the accuracy of the model predictions (Neres et al., 2016).

In this work, we propose two tests on the earthquake recurrence models used to make PSHA studies, one simple, rule-of-thumb type, and the other based on numerical modeling. Both relate the seismic moment release predicted by the models and the

knowledge of lithospheric deformation inferred from dynamic modeling and geological and geodetic observations.

### 4.1 Simple consistency test

For the application of the simple consistency test, the double truncated Gutenberg-Richter law is used, and the seismic strain is converted into a relative velocity between blocks, in mm/year, as in Bird and Kagan (2004) and Mazzotti et al. (2008). The

obtained values should be compared with the convergence rate between the Nubia and Eurasia plates, with a relative velocity of ~4 mm/year (Fernandes et al., 2003). For this purpose, a simplified seismic generation model is used, with seismic strain concentrated on a single fault, representative of the zone under study, as shown in Fig. 5.

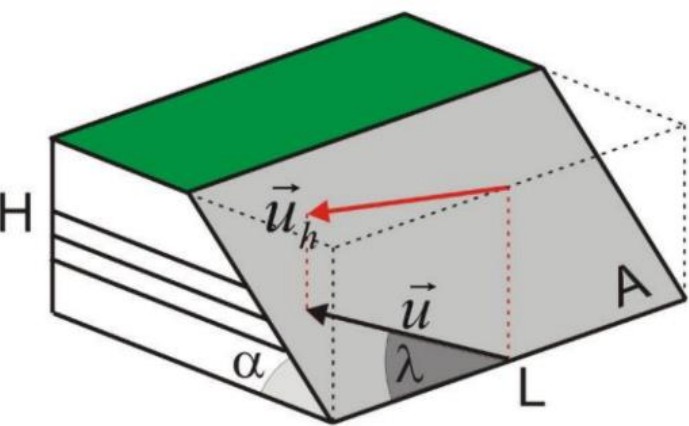

**Figure 5:** Simple earthquake generation model used to calculate the seismic strain rate. Being, A the area of the fault; L the length of the fault; H the thickness of the seismogenic layer and $\alpha$ the dip of fault (Matias and Teves-Costa, 2011a, b).





Considering that all earthquakes occur on a single fault plane with the same mechanism, then the relative velocity between blocks is given by

$\Delta \dot{u}_s = \frac{\sum M_i}{\mu A T}$                                                    (3)

Here $M_i$ is the seismic moment of a single earthquake, $T$ is the period considered and $\mu$ the mean shear modulus for the seismogenic layer. The area of the fault, $A$ (Fig. 5) is determined by its length $L$, the thickness of the seismogenic layer $H$ and the dip of the fault $\alpha$, by

$A = \frac{LH}{\sin\alpha}$                                                             (4)

To apply the simplified test, we consider that the earthquake recurrence models are expressed in moment magnitude, $m_W$ and use the Hanks and Kanamori's law (1979) to convert magnitude into seismic moment

$M_o = 10^{1.5 m_w + 9.1}$                                                    (5)

The velocity along the fault obtained from Eq. (3) must be converted to horizontal velocity by

$\dot{u}_h = \dot{u}\sqrt{(\cos\lambda)^2 + (\sin\lambda\,\cos\alpha)^2}$                              (6)

In the simple test applied in this work, we consider that all seismicity occurs in pure thrust faults, for all generation zones, in which case one has $u_h = u\,\cos\alpha$. In this approximation, the most favorable fault plane dip for thrust faults is 35 ° for the seismogenic zone likely to have generated the 1755 earthquake (e.g. Sørensen et al., 2012), and 55 ° for the Lower Tagus Valley, corresponding to the Continental Rift Boundary class of the work of Bird and Kagan (2004). However, this 55º and 35º dip difference is not relevant since $\sin 35 \cos 35 = \sin 55 \cos 55 = 0.47$,

Thus, the slip rate on the fault due to the accumulated seismic activity, $\Delta \dot{u}_s$, is given by

$\Delta \dot{u}_s = \frac{\sum M_i}{\mu LHT} \sin 35º \cos 35º \quad \rightarrow \quad \Delta \dot{u}_s \approx \frac{1}{2}\frac{\sum M_i}{\mu LHT}$                      (7)

This equation is valid for a perfect seismic coupling, the fraction of frictional sliding that occurs as earthquakes. If the seismic coupling, $c$, is not perfect ($c < 1$), then the slip rate between the blocks should be corrected by this factor,

$\Delta \dot{u}_s \approx \frac{1}{c}\frac{1}{2}\frac{\sum M_i}{\mu LHT}$                                                (8)

The values obtained by the Eq. (7) (for $c = 1$) must, therefore, be considered as the lower limits of the tectonic deformation inferred by the earthquake generation models. In Bird and Kagan (2004) the seismic coupling is very variable for the different plate boundaries investigated, it varies from 0.05, in oceanic rifts in strike-slip faults, to 1.0 in continental convergence regions.

To apply the simplified consistency test to the earthquake generation models presented earlier, it is necessary to assign for each generation zone the following parameters ($\mu, L, H$). While L is taken from the geometry of each source zone, the values


of $\mu$, the mean shear modulus for the seismogenic layer, and the thickness of the seismogenic layer, $H$, must be inferred from our best knowledge on the source areas. For the LTV we consider that it comprises typical continental crust and we used $\mu = 4.0 \times 10^{10} \, Pa$ (e.g. Sørensen et al., 2012). For the 1755 earthquake source zone, we used a typical value for oceanic lithosphere (e.g. Johnston, 1996 and Matias et al., 2013), $\mu = 6.5 \times 10^{10} \, Pa$.

For the value of the brittle thickness of the lithosphere, $H$, we assume that it is 20 km in the LTV where it includes only the
upper and middle continental crust. For the 1755 zone we used H=60 km from Matias et al., 2013. A summary of the parameters used in the application of the simple consistency test is presented in Table 5. The cumulative seismic moment to be used in Eq. (7) is computed from the seismic hazard model investigated.

**Table 5:** Source zone parameters for the simplified sanity check.

| Model | Lower Tagus Valley source zone $\mu = 4.0 \times 10^{10} \, Pa$ | | | 1755 source zone $\mu = 6.5 \times 10^{10} \, Pa$ | | |
|---|---|---|---|---|---|---|
| | $L$ (km) | $H$ (km) | $\alpha$ (º) | $L$ (km) | $H$ (km) | $\alpha$ (º) |
| EC8 | 201 | 20 | 55 | 260 | 60 | 35 |
| ERSTA | 188 | 20 | 55 | 210 | 60 | 35 |
| SHARE | 129 | 20 | 55 | 260 | 60 | 35 |
| QREN | 188 | 20 | 55 | 263 | 60 | 35 |
| SA | 196 | 20 | 55 | 253 | 60 | 35 |
| SB | 270 | 20 | 55 | 209 | 60 | 35 |

To have a perception of the sensitivity of the results to the simple model parameters proposed, which are inversely related,
observing Eq. (7), we see that an increase in the length of the zone ($L$) or the thickness of the layer ($H$), would imply a proportional decrease of the seismic tectonic velocity ($\Delta \dot{u}_s$), and a decrease in these parameters would imply a corresponding increase. The values chosen for each PSHA model, together with c=1, are conservative and so the results obtained should be considered as the lower limits for the tectonic deformation inferred by the earthquake recurrence models.

**4.2 Uncertainties: Logic Tree and Monte-Carlo simulation**

Among the five PSHA models investigated, the Vilanova and Fonseca (2007) is the only one that addresses epistemic uncertainties on the source definition and earthquake recurrence laws using a logic tree (Fig. 4). This results in 32 possible earthquake generation scenarios, each with a fixed probability given by the sequence of branches in the logic tree. We will

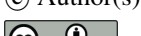



expand the resulting discrete probability distribution into a continuous probability distribution through a Monte-Carlo

simulation (Fig. 6), considering a probability distribution for parameters a and b of the Gutenberg-Richter law. This exercise

will allow to illustrate how the simplified, and complex, sanity checks can be applied to models where the uncertainty in the

DTGR laws is provided. Furthermore, it will help us to choose from the 32 possible scenarios 5 that are representative of 5

classes to be defined by 20% probability intervals on the continuous probability distribution of seismic velocity (Fig. 7). These

5 scenarios will be also investigated by the complex sanity test described below.





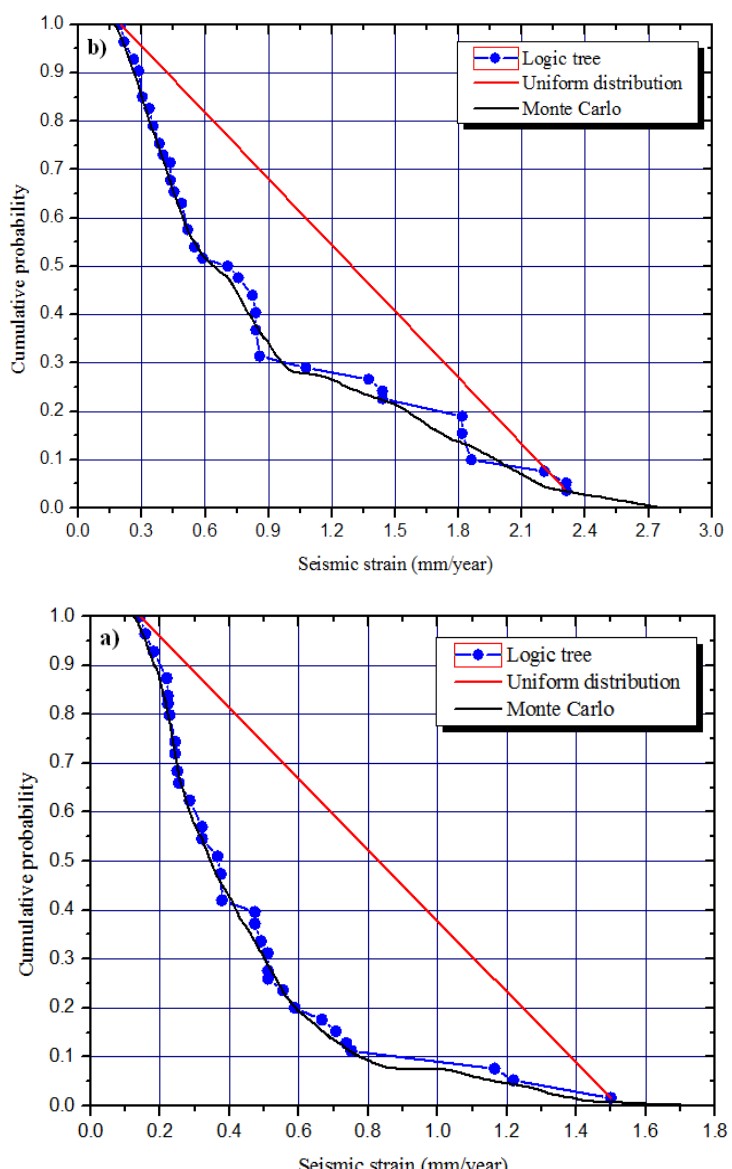

**Figure 6:** Monte-Carlo method applies to Vilanova and Fonseca (2007) Logic Tree models, compared with a uniform probability distribution. a) the LTV source zone. b) the 1755 source zone.

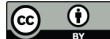

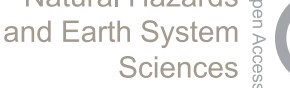

**Figure 7:** Seismic strain and probability for each of the 32 models that result from the logic tree proposal of Vilanova and Fonseca (2007). The scenarios outlined are the ones chosen to be tested with neotectonics modelling. a) the LTV source zone. b) the 1755 source zone.


### 435 **4.3 Complex sanity test: Seismicity Rates with "Long Term Seismicity"**

In slow deforming areas where significant seismic hazard is generated offshore, neotectonic modelling is a very useful tool to infer the long-term deformation rates of the brittle lithosphere. That work was performed for the Africa-Eurasia plate boundary extending from the Gloria Fault to the northern Algerian margin by Neres et al. (2016). These authors improved the broader Mediterranean model (Carafa et al., 2015) compiling an up-to-date simplified tectonic map of the Africa-Iberia plate boundary

and used the code SHELLS (Bird, 1999; Bird et al., 2008), that solves the stress equilibrium equation for a finite element grid, based on vertically integrated and laterally varying lithospheric data, estimated rock strengths and densities, and lateral, basal, and/or internal boundary conditions. SHELLS determines the long-term averages of tectonic strain and motion over many earthquake cycles and outputs results such as horizontal surface velocities, slip rates on faults, strain rates in the continuum elements, and stress orientation. The finite element grid used for the study of Neres et al. (2016) comprised almost equilateral

spherical triangles with about 27 km of side length. More details on the modelling can be found in Neres et al. (2016).

In the present work, the inferred lithospheric deformation, diffuse and concentrated along fault planes, is converted to earthquake generation rates by a modified version of the code "Long Term Seismicity" (Bird and Liu, 2007, Bird et al., 2010) which basic principles are revised below.

Each triangular finite element is subject to horizontal strain rate along the longitude and latitude coordinate system $(\phi, \theta)$ that

is computed from the velocity on its nodes. The vertical strain rate, $\dot{\varepsilon}_{rr}$, is then computed invoking incompressibility

$$\dot{\varepsilon}_{\phi\phi} + \dot{\varepsilon}_{\theta\theta} + \dot{\varepsilon}_{rr} = 0 \qquad (9)$$

Since $\dot{\varepsilon}_{r\phi}$ and $\dot{\varepsilon}_{r\theta}$ should vanish at a shear traction free surface, $\dot{\varepsilon}_{rr}$ is a principal strain rate. Then, the two principal strain rates in the horizontal plane $(\dot{\varepsilon}_{1h} \leq \dot{\varepsilon}_{2h})$, are given by

$$\dot{\varepsilon}_{1h} = \frac{\dot{\varepsilon}_{\phi\phi} + \dot{\varepsilon}_{\theta\theta}}{2} - \sqrt{\dot{\varepsilon}_{\phi\theta}^2 + \left(\dot{\varepsilon}_{\phi\phi} - \dot{\varepsilon}_{\theta\theta}\right)^2/4}$$

$$\dot{\varepsilon}_{2h} = \frac{\dot{\varepsilon}_{\phi\phi} + \dot{\varepsilon}_{\theta\theta}}{2} - \sqrt{\dot{\varepsilon}_{\phi\theta}^2 + \left(\dot{\varepsilon}_{\phi\phi} - \dot{\varepsilon}_{\theta\theta}\right)^2/4} \qquad (10)$$

The three principal strain rates are alternately labelled as an ordered triplet

$$\dot{\varepsilon}_1 \leq \dot{\varepsilon}_2 \leq \dot{\varepsilon}_3 \qquad (11)$$

The seismic moment release rate per unit area, $\dot{M}/A$, for the finite element of area A, is then computed by

$$\frac{\dot{M}}{A} = \langle cz\rangle\mu \begin{cases} 2\dot{\varepsilon}_3; & if\ \dot{\varepsilon}_2<0,\ \ or \\ -2\dot{\varepsilon}_1; & if\ \dot{\varepsilon}_2\geq0 \end{cases} \qquad (12)$$




Here $\mu$ is the average shear modulus and $\langle cz \rangle$ is the coupled thickness, an average of the product of the seismic coupling parameter, $c$, and the depth of the lithosphere above the brittle/ductile transition, z. The seismic moment release equation is based on a kinematic model, where the volume preserving deformation is equivalent to slip on many minor virtual faults falling into (up to) two conjugate sets. The more active conjugate fault set bisects the angles between the principal strain rate axes $\pm\dot{\varepsilon}_1$ and $\pm\dot{\varepsilon}_3$; the less active conjugate fault set bisects the angles between principal strain rate axes $\pm\dot{\varepsilon}_2$ and $\pm\dot{\varepsilon}_3$ (if $\dot{\varepsilon}_2 < 0$)

or between $\pm\dot{\varepsilon}_1$ and $\pm\dot{\varepsilon}_2$ (if $\dot{\varepsilon}_2 \geq 0$). The factor of $\pm 2$ appearing in Eq. (12) is the smallest coefficient possible and comes from the assumption that the virtual fault planes make angles of 45º with the principal strain rate axes (see Appendix of Carafa et al., 2017). This formula also assumes that the strain rates are long-term permanent strain rates, and not elastic (Bird et al., 2010).

When lithospheric deformation is concentrated along fault planes with long-term average slip rate, $\dot{s}$, then, according to Bird

and Liu (2007), the long-term average seismic moment rate on that fault is given by

$$\dot{M} = \iint c\,\mu\,\dot{s}\,da \qquad (13)$$

Where $c$ is the dimensionless seismic coupling, $da$ is an element of fault area, and the integral is over the frictional (potentially seismogenic) portion of the fault surface that lies above the brittle/ductile transition. For large blocks of lithosphere, which do not rotate about horizontal axes (although they may rotate about vertical axes), slip rates hardly vary in the down-dip direction,

allowing for the following approximation (ibid.)

$$\dot{M} \cong \langle cz \rangle \int \mu\sqrt{v_p^2 + (v_o\,\sec(\alpha))^2}\,\csc(\alpha)\,dl \qquad (14)$$

Here $v_p$ is the trace-parallel component of the horizontal relative block velocity vector, $v_o$ is the orthogonal (trace-normal) component of the horizontal relative block velocity vector, $\alpha$ is the dip of the fault and $dl$ is a small step along the length of the fault. The integral is taken on the surface along the trace of the fault. We remark that in both cases, seismic strain from

diffuse or concentrated deformation, one key parameter is the "mean coupled seismogenic thickness", $\langle cz \rangle$. Only the product is relevant in most computations.

In its original form the "Long Term Seismicity" code applies the two hypotheses of the SHIFT model (Seismic Hazard Inferred From Tectonics) proposed by Bird and Liu (2007) and Bird et al. (2010): (a) the calculation of the long-term seismic moment rate in a deformed volume uses the coupled seismogenic thickness defined by the plate boundary closest to the zone; (b)

calculating the earthquake frequency, as well as their magnitude distribution, follows the recurrence law of the closest and most comparable type of plate boundary. For this purpose, the code uses the plate boundary classification proposed by Bird and Kagan (2004) that considered 7 different types of plate boundary: CRB, Continental Rift Boundary; CTF, Continental Transform Fault; CCB, Continental Convergent Boundary; OSR, Oceanic Spreading Ridge; OTF, Oceanic Transform Fault; OCB, Oceanic Convergent Boundary; SUB, Subduction zone. Using the global CMT catalogue, Bird and Kagan in 2004,





estimated the seismicity parameters for each plate boundary type. Instead of using the DTGR recurrence law described above,
these authors used the "Tapered Gutenberg-Richter" law (TGR) which is expressed in terms of seismic moment, not magnitude,

$$\dot{N}(m_T) = \dot{N}^{Comp} \left(\frac{M_T}{M_c}\right)^{-\beta_{tGR}} e^{\left(\frac{M^{Comp}-M_T}{M_c}\right)} \tag{15}$$

$\dot{N}(m_T)$ is the earthquake rate of events with magnitude greater than or equal to the magnitude of interest, $m_T$, with seismic
moment $M_T$. $M_c$ is the seismic moment corresponding to the corner magnitude, $m_c$, which is a magnitude close but not equal

to the maximum magnitude of the DTRG law. The $\beta_{tGR}$ is the asymptotic spectral slope in the low magnitudes of the Tapered
Gutenberg-Richter law and should not be confused with the $\beta$ of the DTRG, the relation between the two is given by

$$\beta = \frac{3}{2}\beta_{tGR}\ln(10) \tag{16}$$

$M^{Comp}$ is the seismic moment corresponding to the magnitude of completeness, $m_{Comp}$, equivalent to the minimum magnitude
defined earlier for the DTGR. $\dot{N}^{Comp}$ is the earthquake rate for events with magnitude greater than or equal to the magnitude

of completeness.

Once the long-term seismic moment rate ($\dot{M}$) of a grid cell is determined, its expected long-term shallow seismicity rate is
obtained in two steps. First the long-term moment rate is divided by the model moment rate (integral of best-fitting "Tapered"
Gutenberg-Richter distribution, $\dot{M}^{CMT}$) of the appropriate Bird and Kagan (2004) CMT subcatalogue and then it is multiplied
by the number of events in that subcatalogue, $\dot{N}^{CMT}$, to determine the rate of earthquakes at the grid cell that will exceed the

threshold magnitude of that subcatalogue,

$$\dot{N}(m > m_T^{CMT}) = \left(\frac{\dot{M}}{\dot{M}^{CMT}}\right) \dot{N}^{CMT} \tag{17}$$

Then the forecasted earthquake rate adjusted to any desired threshold magnitude ($m_T$) using the TGR law is given by,

$$\dot{N}(m > m_T) = \dot{N}(m > m_T^{CMT}) \left(\frac{M(m_T)}{M(m_T^{CMT})}\right)^{-\beta} \exp\left(\frac{M(m_T^{CMT})-M(m_T)}{M(m_c)}\right) \tag{18}$$

The main parameters required to use the "Long Term Seismicity" code to estimate seismic moment release for a particular

zone are: the coupled thickness of seismogenic lithosphere, $\langle cz \rangle$, the shear modulus, $\mu$, the corner magnitude, $m_c$, and the
asymptotic spectral slope, $\beta_{tGR}$, of the TGR law, the number of shallow earthquakes in the catalogue, $N^{Comp}$ and the threshold
magnitude, $m_T$, used in counting those events.

To establish a sanity check on the earthquake source models used for PSHA in W and SW Iberia, a complex tectonic domain
with slow deformation, several modifications had to be done to the main algorithm of "Long Term Seismicity". Firstly, we

introduced the parametrization of zones, instead of plate boundaries. Each finite element cell was identified as belonging to
one source zone. Secondly, instead of using the default plate boundary parameters defined by Bird and Kagan (2004) we





introduced the recurrence parameters of the PSHA models to be investigated. These parameters are given for the source zones, LTV and 1755, respectively, in Tables 6 and 7.

**Table 6:** Earthquake recurrence for the Lower Tagus Valley source zones expressed as DTGR and TGR laws, for the minimum magnitude of 5.0.

| | Double truncated G-R law | | | Tapered G-R law | | | $\dot{M}_0$ (Nm/year) |
|---|---|---|---|---|---|---|---|
| Lower Tagus Valley (LTV) source zone | | | | | | | |
| **Model** | $\lambda$ (m=5.0) | $\beta$ | $m_{max}$ | $\dot{N}^{Comp}$ (EQs/year) | $\beta_{tGR}$ | $m_c$ | |
| EC8 | 0.07 | 1.63 | 7.2 | 0.086 | 0.472 | 6.78 | $1.52 \times 10^{17}$ |
| ERSTA | 0.12 | 1.82 | 7.1 | 0.149 | 0.527 | 6.69 | $1.72 \times 10^{17}$ |
| SHARE$_{min}$ | 0.08 | 2.07 | 7.1 | 0.099 | 0.599 | 6.71 | $8.69 \times 10^{16}$ |
| SHARE$_{med}$ | 0.08 | 2.07 | 7.4 | 0.098 | 0.599 | 7.00 | $1.33 \times 10^{17}$ |
| SHARE$_{max}$ | 0.08 | 2.07 | 7.6 | 0.098 | 0.599 | 7.20 | $1.77 \times 10^{17}$ |
| QREN | 0.01 | 2.03 | 6.87 | 0.012 | 0.588 | 6.47 | $9.12 \times 10^{15}$ |
| SA-CB-RA-a1-max0 | 0.02 | 2.05 | 6.9 | 0.022 | 0.594 | 6.57 | $1.63 \times 10^{16}$ |
| SA-CB-RB-a1-max+ | 0.02 | 2.28 | 7.4 | 0.022 | 0.660 | 7.09 | $2.49 \times 10^{16}$ |
| SA-CB-RB-a2-max0 | 0.03 | 2.28 | 6.9 | 0.034 | 0.660 | 6.58 | $1.92 \times 10^{16}$ |
| SB-CA-RB-a2-max0 | 0.05 | 2.30 | 7.1 | 0.056 | 0.666 | 6.79 | $4.02 \times 10^{16}$ |
| SB-CB-RB-a1-max+ | 0.03 | 2.35 | 7.6 | 0.033 | 0.680 | 7.29 | $4.46 \times 10^{16}$ |






**Table 7:** Earthquake recurrence for the 1755 source zones expressed as DTGR and TGR laws, for the minimum magnitude of 5.0.

| | | | | | | | $\dot{M}_0$ (Nm/year) |
|---|---|---|---|---|---|---|---|
| **1755 Source zone** | | | | | | | |
| | **Double truncated G-R law** | | | **Tapered G-R law** | | | |
| **Model** | $\lambda$ (m=5.0) | $\beta$ | $m_{max}$ | $\dot{N}^{Comp}$ (EQs/year) | $\beta_{tGR}$ | $m_c$ | |
| EC8 | 0.13 | 1.66 | 8.8 | 0.157 | 0.481 | 8.38 | $4.51 \times 10^{18}$ |
| ERSTA | 0.22 | 1.43 | 8.7 | 0.262 | 0.414 | 8.26 | $1.18 \times 10^{19}$ |
| SHARE$_{min}$ | 0.32 | 2.07 | 8.5 | 0.388 | 0.599 | 8.10 | $2.47 \times 10^{18}$ |
| SHARE$_{med}$ | 0.32 | 2.07 | 8.7 | 0.388 | 0.599 | 8.30 | $3.27 \times 10^{18}$ |
| SHARE$_{max}$ | 0.32 | 2.07 | 8.8 | 0.388 | 0.599 | 8.40 | $3.75 \times 10^{18}$ |
| QREN | 0.03 | 2.12 | 8.9 | 0.036 | 0.614 | 8.51 | $4.07 \times 10^{17}$ |
| SA-CA-RA-a2-max+ | 0.09 | 2.37 | 9.0 | 0.099 | 0.686 | 8.69 | $6.34 \times 10^{17}$ |
| SA-CA-RB-a2-max+ | 0.10 | 2.60 | 9.0 | 0.109 | 0.753 | 8.70 | $3.51 \times 10^{17}$ |
| SA-CB-RB-a1-max0 | 0.09 | 2.53 | 8.5 | 0.099 | 0.733 | 8.20 | $2.50 \times 10^{17}$ |
| SB-CA-RA-a1-max+ | 0.12 | 2.16 | 9.0 | 0.133 | 0.625 | 8.68 | $1.45 \times 10^{18}$ |
| SB-CA-RB-a1-max+ | 0.08 | 2.51 | 9.0 | 0.088 | 0.727 | 8.70 | $3.72 \times 10^{17}$ |

However, some preliminary work had to be done to translate the published DTGR law parameters into the TGR law parameters

that are used by "Long Term Seismicity". This task was performed by a non-linear fit, adjusting the corner magnitude and the completeness earthquake rate $\dot{N}^{Comp}$, fixing the asymptotic spectral slope $\beta_{tGR}$ to its theoretical value given by Eq. (16). The quality of the fit was assessed by comparing the total seismic moment release computed for both models, DTGR and fitted TGR. The details of this computation are shown as a supplement.

To facilitate the interpretation of results, all recurrence parameters are scaled to an observation period of 100 years. As regards

the coupled thickness, we consider that seismic coupling was perfect and used 20 and 60 km as seismogenic thickness for LTV and 1755 zones, respectively. The output tectonic seismicity rates are finally compared to observations to check the consistency of the proposed source zones and earthquake recurrence models.

**5. Seismic deformation consistency tests: results and discussion**





**540** **5.1 Simplified test**

This test consisted in computing the fault slip rate that would explain the seismicity from the earthquake recurrence models proposed for PSHA, as described in section 4.1. The slip rates and moment release rates for EC8, ERSTA, SHARE and QREN models considered on the LTV and 1755 source zones are shown on Table 8, and due to the use of a logic tree, as well to simplify, the VF2007 model is shown on Table 9.

**545**

**Table 8:** Simple test: seismic deformation rates in mm/year for the LTV and 1755 source zones considering four earthquake recurrence models.

| | Lower Tagus Valley (LTV) | | 1755 Source zone | |
|---|---|---|---|---|
| **Model** | $\dot{M}$ (Nm/year) | $\Delta\dot{u}_s$ (mm/year) | $\dot{M}$ (Nm/year) | $\Delta\dot{u}_s$ (mm/year) |
| EC8 | $1.52\times10^{17}$ | 0.943 | $4.51\times10^{18}$ | 4.448 |
| ERSTA | $1.72\times10^{17}$ | 1.146 | $1.18\times10^{19}$ | 14.408 |
| $SHARE_{min}$ | $8.69\times10^{16}$ | 0.842 | $2.47\times10^{18}$ | 2.436 |
| $SHARE_{med}$ | $1.33\times10^{17}$ | 1.289 | $3.27\times10^{18}$ | 3.225 |
| $SHARE_{max}$ | $1.77\times10^{17}$ | 1.715 | $3.75\times10^{18}$ | 3.698 |
| QREN | $9.12\times10^{15}$ | 0.061 | $4.07\times10^{17}$ | 0.397 |

**550**





**Table 9:** Simple test: seismic deformation rates in mm/year for the LTV and 1755 source zones, considering Vilanova and Fonseca (2007) 32 models resulting from the logic tree proposal. In bold we identify the recurrence models, and values, that will be checked by the complex sanity test.

| Model | Lower Tagus Valley (LTV) | | | | 1755 Source zone | | | |
|---|---|---|---|---|---|---|---|---|
| | $m_{max}$ | | $m_{max}$+0.5 | | $m_{max}$ | | $m_{max}$+0.5 | |
| | $\dot{M}_0$ (Nm/year) | $\Delta\dot{u}_s$ (mm/year) | $\dot{M}_0$ (Nm/year) | $\Delta\dot{u}_s$ (mm/year) | $\dot{M}_0$ (Nm/year) | $\Delta\dot{u}_s$ (mm/year) | $\dot{M}_0$ (Nm/year) | $\Delta\dot{u}_s$ (mm/year) |
| SA-CA-RA-a1 | $1.42\times10^{16}$ | 0.120 | $2.28\times10^{16}$ | 0.194 | $3.95\times10^{17}$ | 0.412 | $7.50\times10^{17}$ | 0.782 |
| **SA-CA-RA-a2** | $2.88\times10^{16}$ | 0.245 | $6.37\times10^{16}$ | 0.541 | $6.42\times10^{17}$ | 0.670 | $9.81\times10^{17}$ | **0.643** |
| **SA-CB-RA-a1** | $2.04\times10^{16}$ | **0.104** | $3.38\times10^{16}$ | 0.287 | $7.63\times10^{17}$ | 0.796 | $1.65\times10^{18}$ | 1.724 |
| SA-CB-RA-a2 | $4.59\times10^{16}$ | 0.390 | $1.10\times10^{17}$ | 0.932 | $1.31\times10^{18}$ | 1.365 | $2.10\times10^{18}$ | 2.192 |
| SA-CA-RB-a1 | $1.27\times10^{16}$ | 0.109 | $1.97\times10^{16}$ | 0.168 | $1.95\times10^{17}$ | 0.204 | $3.21\times10^{17}$ | 0.335 |
| **SA-CA-RB-a2** | $2.24\times10^{16}$ | 0.191 | $4.41\times10^{16}$ | 0.375 | $3.49\times10^{17}$ | 0.363 | $5.00\times10^{17}$ | **0.356** |
| **SA-CB-RB-a1** | $1.63\times10^{16}$ | 0.139 | $2.57\times10^{16}$ | **0.159** | $2.59\times10^{17}$ | **0.253** | $4.45\times10^{17}$ | 0.464 |
| **SA-CB-RB-a2** | $3.29\times10^{16}$ | **0.123** | $6.76\times10^{16}$ | 0.575 | $4.71\times10^{17}$ | 0.491 | $6.89\times10^{17}$ | 0.718 |
| **SB-CA-RA-a1** | $2.18\times10^{16}$ | 0.135 | $4.26\times10^{16}$ | 0.263 | $7.81\times10^{17}$ | 0.985 | $1.69\times10^{18}$ | **1.781** |
| SB-CA-RA-a2 | $2.00\times10^{16}$ | 0.123 | $4.60\times10^{16}$ | 0.284 | $1.31\times10^{18}$ | 1.652 | $2.10\times10^{18}$ | 2.654 |
| SB-CB-RA-a1 | $2.18\times10^{16}$ | 0.135 | $4.26\times10^{16}$ | 0.263 | $7.63\times10^{17}$ | 0.963 | $1.65\times10^{18}$ | 2.087 |
| SB-CB-RA-a2 | $2.00\times10^{16}$ | 0.123 | $4.60\times10^{16}$ | 0.284 | $1.25\times10^{18}$ | 1.578 | $2.01\times10^{18}$ | 2.534 |
| **SB-CA-RB-a1** | $3.40\times10^{16}$ | 0.210 | $6.00\times10^{16}$ | 0.370 | $2.37\times10^{17}$ | 0.300 | $4.13\times10^{17}$ | **0.456** |
| **SB-CA-RB-a2** | $6.64\times10^{16}$ | **0.186** | $1.35\times10^{17}$ | 0.833 | $3.64\times10^{17}$ | 0.459 | $5.35\times10^{17}$ | 0.675 |
| **SB-CB-RB-a1** | $2.89\times10^{16}$ | 0.178 | $4.99\times10^{16}$ | **0.207** | $1.83\times10^{17}$ | 0.231 | $3.04\times10^{17}$ | 0.385 |
| SB-CB-RB-a2 | $5.30\times10^{16}$ | 0.327 | $1.05\times10^{17}$ | 0.647 | $2.75\times10^{17}$ | 0.347 | $3.97\times10^{17}$ | 0.501 |






As already mentioned, the logic tree proposed by Vilanova and Fonseca (2007) generates 32 possible source zone and recurrence models, which define a discrete probability distribution. This can be converted into a continuous probability distribution considering that the a and b parameters from the proposed recurrence models are affected by uncertainty. Using

Monte-Carlo simulation, the seismic slip rate probability distribution obtained are shown in Fig. 6 for the LTV and 1755 source zones. Splitting the continuous probability distribution in 5 sections 20% wide, then it is possible to select from each section one representative model to be tested against neotectonic modelling. These selected models are graphically displayed in Fig. 7.

The consistency tests provide geodynamic constraints for evaluating PSHA models, allowing to identify suspect models, either

for generating too much or too less earthquakes than predicted. The slip rate on an idealized fault that accumulates all the seismic strain released on that zone generated by the simple test must now be compared with our knowledge of lithospheric deformation and fault slip rates.

The geodynamic setting of W and SW Iberia is dominated by the WNW-ESE oblique dextral convergence between Nubia and Eurasia plates at ~ 4mm/year (Fernandes et al., 2003). Spatial geodesy provides information on how this convergence is

propagated on land causing distributed deformation. Palano et al. (2015) analysis of GNSS (Global Navigation Satellite System) data showed in SW Iberia a gradient in the measured velocities relative to fixed Eurasia, from ~ 2 mm/year in the SE to ~ 0.5 mm/year in the NW. The authors interpreted these results as evidence for a rigid rotation of the Iberian Peninsula around a pole located north of Madrid. Once this rotation is removed, the intraplate deformation is less than 1 mm/year everywhere (ibid.).

Cabral et al. (2017) made a detailed analysis of spatial geodesy data in South Portugal Mainland. These authors identified a shear zone that separates a faster SW domain from a slower NE domain. This shear zone would accommodate part of the velocity gradient already mentioned by Palano et al. (2015) with a relative velocity change of less than 1 mm/year.

Cabral (2012) investigated the intraplate deformation using geological criteria. His synthesis recognizes the existence of several active faults in Portugal Mainland (active in the sense that they show movement evidence in the last 3 Ma) however

with very low slip rates, less than 0.1 mm/year. The most active faults can reach slip rates of 0.5 mm/year. Neotectonic modelling works (Cunha et al., 2012; Neres et al., 2016) confirm these results, as they point to slip rates in intraplate faults less than 0.1 mm/year.

Both space geodesy data and neotectonic modeling results show that the convergence between the African and Eurasian plates, at ~ 4 mm/year, localizes mainly in the ocean domain and must be accommodated there by active faults, some of them already

identified through geological and geophysical studies.

Taking all this information into consideration, we assume 1 mm/year as a reasonable kinematic reference value for the simple sanity check (Tables 8 and 9) for the intraplate Lower Tagus Valley source zone (LTV). For the interplate region where the 1755 earthquake might have been generated, the reference value should be 4 mm/year. Both these values represent upper





limits. Seismic slip rates smaller than 0.1 mm/year or 0.5 mm/year for LTV and 1755 zones, respectively, should also be

considered suspect for they would represent too few seismic deformation. The evaluation of recurrence models from both

sanity tests is deferred to the discussion section.

## 5.2 Seismicity rates compared with "Long Term Seismicity"

According to section 4.3, in slow deforming areas and earthquake offshore domains, neotectonic modelling constrained by

geology, seismicity and geodesy, is a very useful tool to infer lithospheric deformation, as it was done by Neres et al. (2016)

for the W and SW Iberia. Then, lithospheric deformation can be converted to earthquake rates on the source zones used for

PSHA using a modified version of the "Long Term Seismicity" code of Bird and Liu (2007) and Bird et al. (2010). The

earthquake rates computed assuming a perfect coupling can then be compared to observations, either by the number of

earthquakes generated by magnitude classes or by the total moment rate released. This comparison is shown on Tables 10 and

11 for the LTV and 1755 source zones, respectively. It is expressed by the ratio of PSHA recurrence models parameters over

the "Long Term Seismicity" estimate. Keeping all other modelling parameters, this ratio can be interpreted as the seismic

coupling required to fit PSHA recurrence models with neotectonics modelling. These ratios should be considered suspect if

they are close to 1 or greater than 1, meaning that the recurrence models predict more earthquakes and seismic strain than what

would be expected from the neotectonic modelling. On the other hand, very small values of the ratio (smaller than 0.1) should

also be considered suspect because they might indicate a very small and unrealistic seismic coupling.





**Table 10:** Complex test: activity rates in EQs/century for the LTV source zone. Two sets of seismicity parameters were used: 1) LTS obtained with the "Long Term Seismicity" program and 2) PSHA models expressed as Tapered Gutenberg-Richter laws (TGR). Ratio represents the ratio between TGR and LTS. The number of earthquakes consider a catalogue period of 100 years. Ratios suspected to be too high are in bold, and too low in italic. Acceptable ratios are in bold and italic.

| Lower Tagus Valley (LTV) | | | | | | | | |
|---|---|---|---|---|---|---|---|---|
| **Model** | | **Magnitude** | | | | | **Mean Ratio** | $\dot{M}_o$ Nm/century |
| | | **5.0** | **5.5** | **6.0** | **6.5** | **7.0** | | |
| EC8 | LTS | 6.85 | 3.00 | 1.26 | 0.407 | 0.031 | | $1.03\times10^{19}$ |
| | TGR | 8.6 | 3.77 | 1.58 | 0.511 | 0.039 | | $1.29\times10^{19}$ |
| | ratio | **1.255** | **1.257** | **1.254** | **1.256** | **1.258** | **1.256** | **1.248** |
| ERSTA | LTS | 5.22 | 2.07 | 0.773 | 0.203 | 0.007 | | $5.31\times10^{18}$ |
| | TGR | 14.9 | 5.92 | 2.21 | 0.580 | 0.021 | | $1.50\times10^{19}$ |
| | ratio | **2.854** | **2.856** | **2.859** | **2.857** | **2.826** | **2.856** | **2.825** |
| SHARE_min | LTS | 9.45 | 3.32 | 1.09 | 0.261 | 0.010 | | $7.45\times10^{18}$ |
| | TGR | 9.8 | 3.47 | 1.15 | 0.274 | 0.010 | | $7.65\times10^{18}$ |
| | ratio | **1.037** | **1.046** | **1.052** | **1.048** | **1.000** | **1.046** | **1.027** |
| SHARE_med | LTS | 6.16 | 2.18 | 0.753 | 0.231 | 0.036 | | $7.45\times10^{18}$ |
| | TGR | 9.8 | 3.47 | 1.20 | 0.368 | 0.057 | | $1.17\times10^{19}$ |
| | ratio | **1.592** | **1.593** | **1.593** | **1.591** | **1.583** | **1.592** | **1.570** |
| SHARE_max | LTS | 4.65 | 1.65 | 0.577 | 0.191 | 0.045 | | $7.45\times10^{18}$ |
| | TGR | 9.8 | 3.47 | 1.22 | 0.402 | 0.095 | | $1.56\times10^{19}$ |
| | ratio | **2.108** | **2.106** | **2.116** | **2.108** | **2.127** | **2.108** | **2.094** |
| QREN | LTS | 9.51 | 3.35 | 1.03 | 0.150 | $4.81\times10^{-4}$ | | $5.44\times10^{18}$ |
| | TGR | 1.2 | 0.42 | 0.13 | 0.019 | --- | | $6.91\times10^{17}$ |
| | ratio | *0.126* | *0.125* | *0.126* | *0.127* | --- | *0.126* | *0.127* |
| SA-CB-RA-a1-max0 | LTS | 7.28 | 2.55 | 0.807 | 0.151 | $2.61\times10^{-3}$ | | $4.77\times10^{18}$ |
| | TGR | 2.2 | 0.77 | 0.25 | 0.047 | --- | | $1.46\times10^{18}$ |
| | ratio | ***0.302*** | ***0.301*** | ***0.310*** | ***0.311*** | --- | ***0.306*** | ***0.306*** |
| SA-CB-RB-a1-max+ | LTS | 4.73 | 1.51 | 0.467 | 0.134 | 0.023 | | $4.77\times10^{18}$ |
| | TGR | 2.2 | 0.70 | 0.22 | 0.063 | 0.011 | | $2.23\times10^{18}$ |
| | ratio | ***0.465*** | ***0.464*** | ***0.471*** | ***0.470*** | ***0.471*** | ***0.470*** | ***0.468*** |
| SA-CB-RB-a2-max0 | LTS | 8.92 | 2.80 | 0.793 | 0.139 | $2.25\times10^{-3}$ | | $4.77\times10^{18}$ |
| | TGR | 3.4 | 1.07 | 0.31 | 0.052 | --- | | $1.79\times10^{18}$ |
| | ratio | ***0.381*** | ***0.382*** | ***0.391*** | ***0.375*** | --- | ***0.382*** | ***0.375*** |
| SB-CA-RB-a2-max0 | LTS | 9.23 | 2.85 | 0.873 | 0.255 | 0.058 | | $1.08\times10^{19}$ |
| | TGR | 5.6 | 1.76 | 0.53 | 0.12 | 0.007 | | $3.86\times10^{18}$ |
| | ratio | ***0.358*** | ***0.358*** | ***0.355*** | ***0.357*** | ***0.362*** | ***0.358*** | ***0.357*** |
| SB-CB-RB-a1-max+ | LTS | 16.01 | 5.02 | 1.51 | 0.352 | 0.021 | | $1.08\times10^{19}$ |
| | TGR | 3.3 | 1.02 | 0.31 | 0.091 | 0.021 | | $3.76\times10^{18}$ |
| | ratio | ***0.350*** | ***0.351*** | ***0.352*** | ***0.341*** | ***0.339*** | ***0.350*** | ***0.348*** |



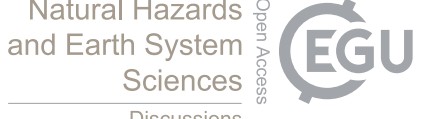

**Table 11:** Complex test: activity rates in EQs/century for the 1755 source zone. Two sets of seismicity parameters were used: 1) LTS obtained with the "Long Term Seismicity" program and 2) PSHA models expressed as Tapered Gutenberg-Richter laws (TGR). Ratio represents the ratio between TGR and LTS. The number of earthquakes consider a catalogue period of 100 years. Ratios suspected to be too high are in bold, and too low in italic. Acceptable ratios are in bold and italic.

| Model | | Magnitude 5.0 | 5.5 | 6.0 | 6.5 | 7.0 | 7.5 | 8.0 | 8.5 | Mean Ratio | $\dot{M}_o$ Nm/century |
|---|---|---|---|---|---|---|---|---|---|---|---|
| EC8 | LTS | 13.6 | 5.91 | 2.58 | 1.12 | 0.485 | 0.203 | 0.071 | $8.9\times10^{-3}$ | | $3.52\times10^{20}$ |
| | TGR | 15.7 | 6.85 | 2.98 | 1.30 | 0.563 | 0.236 | 0.082 | 0.010 | | $4.04\times10^{20}$ |
| | ratio | **1.154** | **1.159** | **1.155** | **1.161** | **1.161** | **1.163** | **1.155** | **1.124** | **1.157** | **1.148** |
| ERSTA | LTS | 6.69 | 3.27 | 1.60 | 0.782 | 0.378 | 0.174 | 0.061 | $4.5\times10^{-3}$ | | $2.66\times10^{20}$ |
| | TGR | 26.2 | 12.8 | 6.27 | 3.06 | 1.48 | 0.683 | 0.239 | 0.018 | | $1.03\times10^{21}$ |
| | ratio | **3.916** | **3.914** | **3.919** | **3.913** | **3.915** | **3.925** | **3.918** | **4.00** | **3.917** | **3.872** |
| SHARE$_{min}$ | LTS | 59.7 | 21.2 | 7.53 | 2.67 | 0.931 | 0.298 | 0.059 | $8.0\times10^{-4}$ | | $3.41\times10^{20}$ |
| | TGR | 38.8 | 13.8 | 4.89 | 1.732 | 0.604 | 0.194 | 0.038 | $5.2\times10^{-4}$ | | $2.20\times10^{20}$ |
| | ratio | *0.650* | *0.651* | *0.649* | *0.649* | *0.649* | *0.651* | *0.644* | *0.650* | *0.650* | *0.645* |
| SHARE$_{med}$ | LTS | 45.2 | 16.1 | 5.71 | 2.03 | 0.713 | 0.241 | 0.064 | $4.4\times10^{-3}$ | | $3.41\times10^{20}$ |
| | TGR | 38.8 | 13.8 | 4.89 | 1.736 | 0.611 | 0.206 | 0.055 | 0.004 | | $2.90\times10^{20}$ |
| | ratio | *0.858* | *0.857* | *0.856* | *0.855* | *0.857* | *0.855* | *0.859* | *0.909* | *0.857* | *0.850* |
| SHARE$_{max}$ | LTS | 39.2 | 13.9 | 4.95 | 1.76 | 0.621 | 0.213 | 0.061 | $6.8\times10^{-3}$ | | $3.41\times10^{20}$ |
| | TGR | 38.8 | 13.8 | 4.89 | 1.737 | 0.613 | 0.230 | 0.061 | 0.007 | | $3.33\times10^{20}$ |
| | ratio | **0.990** | **0.993** | **0.988** | **0.987** | **0.987** | **1.080** | **1.00** | **1.029** | **0.991** | **0.977** |
| QREN | LTS | 59.7 | 20.7 | 7.16 | 2.48 | 0.854 | 0.288 | 0.087 | 0.014 | | $5.19\times10^{20}$ |
| | TGR | 3.6 | 1.25 | 0.43 | 0.15 | 0.052 | 0.017 | 0.005 | $8.2\times10^{-4}$ | | $3.13\times10^{19}$ |
| | ratio | *0.060* | *0.060* | *0.060* | *0.060* | *0.061* | *0.059* | *0.057* | *0.059* | *0.060* | *0.060* |
| SA-CA-RA-a2-max+ | LTS | 86.5 | 26.5 | 8.09 | 2.47 | 0.755 | 0.228 | 0.065 | 0.013 | | $4.73\times10^{20}$ |
| | TGR | 9.9 | 3.03 | 0.93 | 0.283 | 0.086 | 0.026 | 0.007 | $1.47\times10^{-3}$ | | $5.36\times10^{19}$ |
| | ratio | *0.114* | *0.114* | *0.115* | *0.115* | *0.114* | *0.114* | *0.108* | *0.113* | *0.114* | *0.113* |
| SA-CA-RB-a2-max+ | LTS | 158.7 | 43.2 | 11.8 | 3.21 | 0.872 | 0.234 | 0.059 | 0.011 | | $4.73\times10^{20}$ |
| | TGR | 10.9 | 3.0 | 0.810 | 0.221 | 0.060 | 0.016 | 0.004 | $7.4\times10^{-4}$ | | $3.22\times10^{19}$ |
| | ratio | *0.069* | *0.069* | *0.069* | *0.069* | *0.069* | *0.068* | *0.068* | *0.067* | *0.069* | *0.068* |
| SA-CB-RB-a1-max0 | LTS | 214.1 | 60.4 | 17.0 | 4.79 | 1.33 | 0.349 | 0.065 | $1.8\times10^{-3}$ | | $4.73\times10^{20}$ |
| | TGR | 9.9 | 2.79 | 0.79 | 0.22 | 0.062 | 0.016 | 0.003 | $8.0\times10^{-5}$ | | $2.17\times10^{19}$ |
| | ratio | *0.046* | *0.046* | *0.046* | *0.046* | *0.047* | *0.046* | *0.046* | *0.044* | *0.046* | *0.046* |
| SB-CA-RA-a1-max+ | LTS | 26.6 | 9.05 | 3.08 | 1.04 | 0.354 | 0.119 | 0.037 | $8.15\times10^{-3}$ | | $2.59\times10^{20}$ |
| | TGR | 13.3 | 4.52 | 1.534 | 0.521 | 0.176 | 0.059 | 0.019 | $4.05\times10^{-3}$ | | $1.28\times10^{20}$ |
| | ratio | *0.500* | *0.499* | *0.498* | *0.501* | *0.497* | *0.496* | *0.514* | *0.497* | *0.499* | *0.494* |
| SB-CA-RB-a1-max+ | LTS | 68.7 | 19.6 | 5.58 | 1.59 | 0.451 | 0.127 | 0.034 | $6.3\times10^{-3}$ | | $2.59\times10^{20}$ |
| | TGR | 8.8 | 2.51 | 0.72 | 0.204 | 0.058 | 0.016 | 0.004 | $8.2\times10^{-4}$ | | $3.27\times10^{19}$ |
| | ratio | *0.128* | *0.128* | *0.129* | *0.128* | *0.129* | *0.126* | *0.118* | *0.130* | *0.128* | *0.126* |





We present in Table 12 a summary of the sanity tests (simple and complex) performed on the 11 PSHA earthquake generation
models, expressed as seismic slip rates ($\Delta\dot{u}_s$) on a single fault and average ratios between earthquake generation (**Mean Ratio LTS**) and seismic moment release ratios (**Ratio $\dot{M}_o$**) computed by neotectonic modelling with the "Long Term Seismicity" algorithm.

**Table 12:** Summary of the simple ($\Delta\dot{u}_s$) and complex (Ratio $\dot{M}_o$) sanity tests applied to PSHA earthquake generation models. Bold indicates that PSHA model generates too much earthquakes, while italic shows that the model generates too little earthquakes, according to criteria defined in the text. In bold and italic, we show acceptable values.

| Lower Tagus Valley (LTV) | | | | 1755 Source Zone | | | |
|---|---|---|---|---|---|---|---|
| Model | $\Delta\dot{u}_s$ mm/year | Mean Ratio LTS | Ratio $\dot{M}_o$ | Model | $\Delta\dot{u}_s$ mm/year | Mean Ratio LTS | Ratio $\dot{M}_o$ |
| EC8 | *0.943* | **1.256** | **1.248** | EC8 | *4.448* | **1.157** | **1.148** |
| ERSTA | **1.146** | **2.856** | **2.825** | ERSTA | **14.408** | **3.917** | **3.872** |
| SHARE_min | *0.842* | **1.046** | **1.027** | SHARE_min | *2.436* | *0.650* | *0.645* |
| SHARE_med | **1.289** | **1.592** | **1.570** | SHARE_med | *3.225* | *0.857* | *0.850* |
| SHARE_max | **1.715** | **2.108** | **2.094** | SHARE_max | *3.698* | 0.991 | 0.977 |
| QREN | *0.061* | *0.126* | *0.127* | QREN | *0.397* | *0.060* | *0.060* |
| SA-CB-RA-a1-max0 | *0.104* | *0.306* | *0.306* | SA-CA-RA-a2-max+ | *0.643* | *0.114* | *0.113* |
| SA-CB-RB-a1-max+ | *0.159* | *0.470* | *0.468* | SA-CA-RB-a2-max+ | *0.356* | *0.069* | *0.068* |
| SA-CB-RB-a2-max0 | *0.123* | *0.382* | *0.375* | SA-CB-RB-a1-max0 | *0.253* | *0.046* | *0.046* |
| SB-CA-RB-a2-max0 | *0.186* | *0.358* | *0.357* | SB-CA-RA-a1-max+ | *1.781* | *0.499* | *0.494* |
| SB-CB-RB-a1-max+ | *0.207* | *0.350* | *0.348* | SB-CA-RB-a1-max+ | *0.456* | *0.128* | *0.126* |


### 5.3 Joint evaluation of earthquake generation models

The most realistic test, the comparison between PSHA earthquake generation models and neotectonic modelling for the LTV source zone, clearly identifies three models with excessive earthquake generation, EC8, ERSTA and SHARE_min, SHARE_max





and one suspect, SHARE$_{med}$. These three models were also considered suspect by the simplified consistency test. The QREN
model shows too little earthquake generation, confirmed by both sanity checks, simple and complex. All 5 VF2007 earthquake
generation models studied satisfy both consistency tests.

Considering the 1755 source zone, only 3 PSHA scenarios pass both sanity tests, SHARE$_{min}$, SHARE$_{med}$ and one of VF2002
models. The complex consistency test identifies scenarios EC8 and ERSTA as being clearly excessive in earthquake
generation, while SHARE$_{max}$ may be considered suspect. Models that generate too few seismic activity, or being suspect for
that, are QREN and 4 of the 5 VF2007 models investigated. In this source zone, the simplified test confirms 9 of the 11
evaluations done by the complex test, disagreeing the diagnostic on one of VF2007 models and SHARE$_{max}$.

All considered, despite its simplicity, we remark that the simplified consistency check evaluation agrees with the more
elaborate one on 20 out of the 22 PSHA earthquake generation models. While in the LTV source zone, an intraplate domain,
6/11 PSHA models fail the tests or are considered suspect, in the 1755 source zone, an interplate domain, the score is lower,
only 4/11 models pass the complex sanity check. The poor performance of PSHA models in the source areas that include the
large 1755 earthquake may be explained by the poor geological control used on its definition, a diffuse plate boundary with
large identified geological faults with mostly unknown activity.

Revisiting Fig. 6 and 7 where all 32 VF2007 models are analysed by the simple sanity test, we see that for the LTV source
zone, three models exceed 1 mm/year seismic slip rates and must be considered suspect. Using Monte-Carlo simulation to
account for uncertainties in the DTGR law parameters, 10 % of those models generated earthquakes in excess. Differently, for
the 1755 source zone, 40 % of the investigated models should be considered suspect for generating too little earthquakes.

It may be argued that the reference values used to identify PSHA models as suspect or unrealistic depend on a few critical
parameters that were subjectively defined in this work, like the thickness of the brittle lithosphere, the seismic coupling, the
dip and length of the typical fault. The seismic moment release and earthquake generation rates are linear functions of these
parameters, directly or inversely related, and it will be a simple algebraic exercise to compute other reference values suiting
the researcher preferences.

It was not the purpose of this work to propose a quantitative methodology to evaluate the likelihood of the PSHA earthquake
generation models, like the N-test proposed by Schorlemmer et al. (2007) and applied, for example, by Carafa et al. (2018) to
the Calabrian Arc observed seismicity rates. Earthquake catalogues in Slow Deforming Regions are mostly incomplete, have
large return periods and the maximum magnitude is usually determined by subjective and/or geological criteria with large
uncertainties. Instead, the consistency between earthquake generation models from PSHA and tectonic deformation converted
into seismic moment release rates with the "Long Term Seismicity" algorithm was expressed in two ways: i) by the ratio of
seismic activity for different magnitude classes and; ii)  by the ratio between seismic moment release rates, which can be
interpreted as the mean seismic coupling coefficient needed for perfect consistency. The results obtained show that both
indicators result in the same classification of the PSHA models.





## 6. Conclusions

Probabilistic Seismic Hazard Assessment studies are commonly used tools for land use management and for the establishment of building codes. In areas of slow deformation and/or where the seismogenic structures cannot be clearly related to instrumental and historical seismicity (e.g. in offshore domains), researchers use the concept of area source to characterize the earthquake generation process. Given the several subjective judgements that are needed to establish these zones and to define the earthquake recurrence laws, it is common that different authors reach different PSHA results, which cannot be ascribed only to the choice of the ground motion prediction equations. We proposed a methodology to objectively evaluate the chosen earthquake generation models, by comparing the seismic moment release with the knowledge on tectonic deformation, by space geodesy and geology or by neotectonic modelling.

One of the areas investigated in W and SW Iberia, a typical slow deforming region, was the Lower Tagus Valley, an area with very high seismic risk since it includes Lisbon and has a large concentration of population, services, and industry. This is an intraplate domain where space geodesy hints at a strain rate smaller than 1 mm/year. The other source area chosen is offshore SW Iberia, an interplate setting, where several studies locate the source of the 1$^{st}$ November 1755 destructive earthquake.

All PSHA studies considered use the double truncated Gutenberg-Richter law (or other relation that can be made equivalent to this one) to express the earthquake recurrence. These laws are translated into earthquake generation rates for magnitude classes and seismic moment release rates, if the moment magnitude is used. Seismic moment release rate can be easily translated into one slip rate on a single fault, considered as representative of the tectonic deformation of the area, as in Mazzotti and Adams (2005). This is the basis for the first sanity check proposed, the simple one, where seismic slip rate is compared to kinematic knowledge from space geodesy and geology.

In the absence of space geodetic and reliable geological observations, as in offshore domains, lithospheric deformation can be modelled by numerical methods, with results validated by comparison with geological, geodetic, and seismic observations. The inferred lithospheric strain can be converted to earthquake and seismic moment release rates using the "Long Term Seismicity" algorithm. The second sanity test proposed compares the PSHA earthquake generation models with the ones from neotectonic modelling. Both tests can be used to assess the consistency of PSHA models. The simple test provided the same classification as the complex test on 20 of the 22 scrutinized models.

Today PSHA studies deal with epistemic and aleatoric uncertainties using logic-tree and Monte-Carlo simulations, which result in a large number of earthquake generation models to be compounded. We have shown, by the example analysed, that these models span 10 fold seismic moment release ratios which may be considered unrealistic for the earthquake generation zones being investigated.



We suggest that the consistency tests proposed in this work should be part of the PSHA studies in regions of slow deformation and/or with offshore complex earthquake generation mechanisms. In the examples investigated only area earthquake source zones were tested but the methodology can be easily extended to PSHA studies that use more detailed source parametrization (e.g. Rivas-Medina et al., 2018). When logic tree and/or Monte-Carlo methods are used to characterize the epistemic and aleatoric uncertainties in PSHA, it is recommended that the outcome of the sanity tests should be used on a weighing scheme, as it is already used in probabilistic tsunami hazard assessment (Davies et al., 2017, Basili et al., 2019).

## 7. Acknowledgements

This publication was supported by FCT project No. UIDB/50019/2020 - IDL.

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
