# Peer review of "A sanity check for earthquake recurrence models used in PSHA of slow deforming regions: the case of SW Iberia"

_Natural Hazards and Earth System Sciences, 2020_

## Referee Comment (RC1) · Roberto Cabieces (Referee) · 17 Mar 2021

Title: A sanity check for earthquake recurrence models used in PSHA of slow deforming regions: the case of SW Iberia Author(s): Margarida Ramalho, Luis Matias, Marta Neres, Michele M. C. Carafa, Alexandra Carvalho, and Paula Teves-Costa

General comments:

The authors show very promising results in the evaluation of earthquake generation models using two different methods. The methodologies have been applied and compared in SW Iberia which is one of the most seismic risk zones.

The manuscript is readable and well structured. I just have two main concerns. The first one is that the methodology section can be corrected to boost the focus of the authors specific contributions (see details below). My second remark is about the conclusions section. Please rewrite this section, it is needed to summarize the most important results of your paper. From my point of view is still difficult to rapidly retrieve the main contributions of your work in the actual conclusion section.

Specific Comments

1. In the abstract, please remove the last sentence. It is better not to include in this section a conclusion, just leave some remarkable results.

2. Line 203 "mentioned above do not consider any uncertainity". Please rewrite this sentence being more precise in "any uncertainity". It could be better to explain what uncertainties you are referring to. To me this sentence is too general and/or strong.

3. Simple consistency test:

It is needed to reduce this section, some content are trivial and can be moved to the supplementary material.

a. Please, remove the content from line 382 (The area....) to line 396 and move it to the supplementary material. b. Line 400, Eq 7 »> Eq 8

4. Uncertainties: Logic Tree and Monte-Carlo simulation

a. It is necessary to include a reference or some brief explanation about how you have implemented the Monte-Carlo simulation (i.e. parametrisation). b. Line 425 (This exercise → This simulation)

5. Complex sanity test_ Seismicity Rates with "Long Term Seismicity"

a. Line 448 (are revised below → see Supplementary Material) b. Please move from line 449 to 481 to the supplementary material. This will be help that the reader focus on your actual work and the section be more organised separating "your work" and "the

fundamental basis". c. Line 482, replace code by algorithm

6. Seismicity rates compared with "Long Term Seismicity"

a. No necessary to include the sentence from line 652 to 653. It doesn't add significant information.

Technical corrections

Figure 1.

a) I suggest that the maps have colorbars. b) Might be you can change the color pallet to highlight the faults. Now, the faults are in white and is difficult to distinguish it. c) I don't think you need an Europe map. You can integrate the rest of the figures in just one figures.

Figure 7.

I suggest that you place a legend with the name of the models and each model be a coloured dot , instead of set the name in the figure.

Tables.

In general, please explain what each column stands for, even though you have already written in the main text.

I hope this helps to improve the authors paper,

Roberto Cabieces

---

## Referee Comment (RC2) · Robert J. Geller (Referee) · 19 Mar 2021

Referee: Robert Geller (anonymity waived)

This paper is headed in the right direction, but only makes a tiny amount of progress. The key issue is stated in very hesitant terms in lines 43-45, namely: "One of the evolutions suffered by PSHA studies and now recognized as essential, is the evaluation of the uncertainties on the results (e.g. Frankel, 2004, Stein et al., 2012, Mulargia et al., 2017), being a consequence of our incomplete knowledge of the earthquake generation and propagation mechanisms." The question though is whether this uncertainty

can be dealt with within the PSHA framework (this seems to be the authors' position) or whether the entire PSHA framework is fundamentally flawed and should be abandoned (this is the position of Mulargia et al., 2017, with which, as a co-author, I agree with). If one agrees there is, at present, no scientific basis for selecting the earthquakes to be fed into PHSA as the input, then the output of PSHA will just be a bunch of numbers, with no physical validity. I'd like to see the authors confront and discuss this issue head-on in their revised paper.

Using the published hazard maps to conduct a "sanity check" is one way to show the existence of problems and inconsistencies, and I support the eventual publication of this work, but it seems to me that this was a fully expectable result, given that (as I understand it) the hazard maps were constructed by assuming that future seismic activity would be the same as past seismic activity.

Seth Stein wrote a book about this problem "Disaster Deferred: A New View of Earthquake Hazards in the New Madrid Seismic Zone" (2010, Columbia University Press). You can find references to some of his papers on New Madrid in that book. I think the Frankel (2004) paper you cite (lines 762-763) is basically a reply to arguments by Stein. It might be useful for you to discuss that controversy in more detail in your paper, as the basic issue seems to be similar to the one involved in hazard maps for Portugal.

It will take a long time for the hazard estimation community to figure out how best to deal with problems of the type discussed in this paper. I hope the authors' revision is done in a way that they address the underlying issues in more depth, and more profoundly.

---

## Author Comment (AC1) · 30 Apr 2021

**Author comments on "A sanity check for earthquake recurrence models used in PSHA of slow deforming regions: the case of SW Iberia" by Margarida Ramalho et al.**

**Title: A sanity check for earthquake recurrence models used in PSHA of slow deforming regions: the case of SW Iberia**
**Author(s): Margarida Ramalho, Luis Matias, Marta Neres, Michele M. C. Carafa, Alexandra Carvalho, and Paula Teves-Costa**

**Interactive comment by Roberto Cabieces (Referee)**

rcabdia@gmail.com

**General comments:**

The authors show very promising results in the evaluation of earthquake generation models using two different methods. The methodologies have been applied and compared in SW Iberia which is one of the most seismic risk zones.

The manuscript is readable and well structured. I just have two main concerns. The first one is that the methodology section can be corrected to boost the focus of the authors specific contributions (see details below). My second remark is about the conclusions section. Please rewrite this section, it is needed to summarize the most important results of your paper. From my point of view is still difficult to rapidly retrieve the main contributions of your work in the actual conclusion section.

R: We thank the reviewer for his constructive comments. We modified the methodology section following the detailed comments and suggestions. We also rewrote the conclusions focusing on the most important results, namely:

1. *That we proposed a methodology to objectively evaluate the chosen earthquake generation models, by comparing the seismic moment release with the knowledge on tectonic deformation, by space geodesy and geology or by neotectonic modelling.*

2. *The basis for the first sanity check proposed, the simple one, where seismic slip rate is compared to kinematic knowledge from space geodesy and geology. In the absence of space geodetic and reliable geological observations, as in offshore domains, lithospheric deformation can be modelled by numerical methods, with results validated by comparison with geological, geodetic, and seismic observations. The inferred lithospheric strain can be converted to earthquake and seismic moment release rates using the "Long Term Seismicity" algorithm. The second sanity test proposed compares the PSHA earthquake generation models with the ones from neotectonic modelling. Both tests can be used to assess the consistency of PSHA models. The simple test provided the same classification as the complex test on 20 of the 22 scrutinized models.*

3. *We suggest that the consistency tests proposed in this work should be part of the PSHA studies in regions of slow deformation and/or with offshore complex earthquake generation mechanisms. In the examples investigated only area earthquake source zones were tested but the methodology can be easily extended to PSHA studies that use more detailed source parametrization (e.g. Rivas-Medina et al., 2018). When logic tree and/or Monte-Carlo methods are used to characterize the epistemic and aleatoric uncertainties in PSHA, it is recommended that the outcome of the sanity tests should be used on a weighing scheme, as it is already used in probabilistic tsunami hazard assessment (Davies et al., 2017, Basili et al., 2019).*

**Specific Comments**

**1.** In the abstract, please remove the last sentence. It is better not to include in this section a conclusion, just leave some remarkable results.

R: We reformulated the last sentence in order to integrate it as a result from this study:

*"Results show that some of the earthquake source models should be considered as suspicious, given their high/low moment release when compared to the expected values from GNSS observations or neotectonic modelling. This analysis allowed for a downgrade of the weight of poorly compliant models in the PSHA analysis, thus for a more realistic hazard assessment, and can be integrated in other studies of similar settings."*

**2.** Line 203 "mentioned above do not consider any uncertainty". Please rewrite this sentence being more precise in "any uncertainty". It could be better to explain what uncertainties you are referring to. To me this sentence is too general and/or strong.

R: We follow the reviewer's suggestion and changed "any uncertainty" to "the uncertainty", obtaining a more precise sentence.

*"Some of the PSHA studies mentioned above do not consider* the *uncertainty in the definition of the earthquake generation model (zoning and earthquake recurrence), others consider that different models can be accepted with some weights, with uncertainty assessed by the logic tree method (e.g. Vilanova and Fonseca, 2007, Woessner et al., 2015), while others also consider that the DTGR parameters have a statistical uncertainty that is assessed by Monte-Carlo simulations (e.g. IGN-UPM, 2013, Gaspar-Escribano et al., 2015)."*

**3.** Simple consistency test:

It is needed to reduce this section, some content are trivial and can be moved to the supplementary material.

a. Please, remove the content from line 382 (The area. . ..) to line 396 and move it to the supplementary material.

R: We agree that this suggestion will result in a clearer version of the manuscript and proceeded accordingly.

b. Line 400, Eq 7 »> Eq 8

R: Corrected according to the resulting numeration in the revised version.

**4.** Uncertainties: Logic Tree and Monte-Carlo simulation

a. It is necessary to include a reference or some brief explanation about how you have implemented the Monte-Carlo simulation (i.e. parametrization).

R: We acknowledge that further information could be given on the applied method and add in the revised version a sentence to clarify it:

*Among the five PSHA models investigated, the Vilanova and Fonseca (2007) is the only one that addresses epistemic uncertainties on the source definition and earthquake recurrence laws using a logic tree (Fig. 4). This results in 32 possible earthquake generation scenarios, each with a fixed probability given by the sequence of branches in the logic tree. Here, we expand the resulting discrete probability distribution into a continuous probability distribution through a Monte-Carlo simulation (Fig. 6), considering a probability distribution for parameters a and b of the Gutenberg-Richter law.* **The Monte-Carlo simulation is implemented to address aleatory uncertainties, to evaluate the variability of the compose set of parameters for which the different characteristics of each scenario is selected stochastically according to its uncertainty distribution (Gaspar-Escribano et al., 2015).** *This exercise will allow to illustrate how the simplified, and complex, sanity checks can be applied to models where the uncertainty in the DTGR laws is provided. Furthermore, it will help us to choose from the 32 possible scenarios 5 that are representative of 5 classes to be defined by 20% probability intervals on the continuous probability distribution of seismic velocity (Fig. 7). These 5 scenarios will be also investigated by the complex sanity test described below.*

b. Line 425 (This exercise → This simulation)

R: Agree. Changed.

**5.** Complex sanity test_ Seismicity Rates with "Long Term Seismicity"

a. Line 448 (are revised below → see Supplementary Material) b. Please move from line 449 to 481 to the supplementary material. This will be help that the reader focus on your actual work and the section be more organized separating "your work" and "the fundamental basis".

R: We agree that this suggestion will result in a clearer version of the manuscript and proceeded accordingly.

c. Line 482, replace code by algorithm

R: Changed.

**6.** Seismicity rates compared with "Long Term Seismicity"

a. No necessary to include the sentence from line 652 to 653. It doesn't add significant information.

R: We agree to remove this sentence.

**Technical corrections**

**Figure 1.**

a) I suggest that the maps have colorbars. b) Might be you can change the color pallet to highlight the faults. Now, the faults are in white and is difficult to distinguish it. c) I don't think you need an Europe map. You can integrate the rest of the figures in just one figures.

R: a) We add a colorbar as shown in the new figure below. b) At this stage we are not able to change the color of the faults. c) We prefer to leave the different parts of the figure, to better illustrate the different aspects and scales of the geodynamic setting.

[Figure]

**Figure 7.**

I suggest that you place a legend with the name of the models and each model be a coloured dot, instead of set the name in the figure.

R: We agree that this will improve the readability of the figure. We are implementing these changes, which will be included in the new version of the manuscript.

**Tables.**

In general, please explain what each column stands for, even though you have already written in the main text.

R: We will implement that, in agreement with this comment, because it is essential that the tables legends contain all the information in it.

We agree and will proceed accordingly for the new version of the manuscript.

---

## Author Comment (AC2) · 30 Apr 2021

**Author comments on "A sanity check for earthquake recurrence models used in PSHA of slow deforming regions: the case of SW Iberia" by Margarida Ramalho et al.**

**Title: A sanity check for earthquake recurrence models used in PSHA of slow deforming regions: the case of SW Iberia**
**Author(s): Margarida Ramalho, Luis Matias, Marta Neres, Michele M. C. Carafa, Alexandra Carvalho, and Paula Teves-Costa**

**Interactive comment by Robert J. Geller (Referee)**

bob@eps.s.u-tokyo.ac.jp

Referee: Robert Geller (anonymity waived)

**RC2 - 1.** This paper is headed in the right direction, but only makes a tiny amount of progress. The key issue is stated in very hesitant terms in lines 43-45, namely: "One of the evolutions suffered by PSHA studies and now recognized as essential, is the evaluation of the uncertainties on the results (e.g. Frankel, 2004, Stein et al., 2012, Mulargia et al., 2017), being a consequence of our incomplete knowledge of the earthquake generation and propagation mechanisms." The question though is whether this uncertainty can be dealt with within the PSHA framework (this seems to be the authors' position) or whether the entire PSHA framework is fundamentally flawed and should be abandoned (this is the position of Mulargia et al., 2017, with which, as a co-author, I agree with). If one agrees there is, at present, no scientific basis for selecting the earthquakes to be fed into PHSA as the input, then the output of PSHA will just be a bunch of numbers, with no physical validity. I'd like to see the authors confront and discuss this issue head-on in their revised paper.

R: We agree that the PSHA has its limitations, but it is still used mainly for several seismic hazard maps. For this reason, given the societal relevance of PSHA, in our work we propose a sanity check to score the existing models in a slow-deforming region.

Following this thought, we decided to better express our opinion adding this sentence to the paper: line 43 "As written before, despite PSHA limitations and flaws, it is the used method by society and scientific community, and here is this paper contribution to the topic.".

**RC2 – 2.** Using the published hazard maps to conduct a "sanity check" is one way to show the existence of problems and inconsistencies, and I support the eventual publication of this work, but it seems to me that this was a fully expectable result, given that (as I understand it) the hazard maps were constructed by assuming that future seismic activity would be the same as past seismic activity.

R: For the specific case of our study area, we focused on different available models relying on the classical PSHA approach of source zonation because there isn't any published model (using alternative techniques to PSHA) to be scored with the neotectonic one. However, in this paper, we don't discuss PSHA models themselves, but we concentrate on the earthquake generation models used to build them. PSHA map differences may also result from the choice of the Ground Motion Equations that are not discussed at all. In fact, given a set of different PSHA models, it is not easy to assess the causes for such differences, a relevant question not addressed in the paper.

In Slow Deforming Regions (as it is our study area), assuming time ergodicity is the usual approach and indeed it is used in the investigated earthquake generation models. At this stage of our research, we are still not entirely convinced that time ergodicity directly affects differences in hazard maps, as assumed by the referee (if we understood this point correctly).

We think that the novelty of our approach rests in determining the earthquake rates from a neotectonics model, which is largely independent of the past-seismicity spatial distribution and mainly relies on long-term slip rates off-fault deformation. This method is more coherent with the expected long-term tectonic evolution of the study region. More, it doesn't assume that the

future seismicity follows the spatial pattern of the available seismic catalog. In any case, we think that the scope of the paper is well expressed in the introduction, section 5.3 and conclusions.

To better clarify this issue in the paper: in figure 2 "*These maps show the inconsistency of PSHA models, as two options for the chosen models.*".

**RC2 - 3.** Seth Stein wrote a book about this problem "Disaster Deferred: A New View of Earthquake Hazards in the New Madrid Seismic Zone" (2010, Columbia University Press). You can find references to some of his papers on New Madrid in that book. I think the Frankel (2004) paper you cite (lines 762-763) is basically a reply to arguments by Stein. It might be useful for you to discuss that controversy in more detail in your paper, as the basic issue seems to be similar to the one involved in hazard maps for Portugal.

It will take a long time for the hazard estimation community to figure out how best to deal with problems of the type discussed in this paper. I hope the authors' revision is done in a way that they address the underlying issues in more depth, and more profoundly.

R: We quote the work by Frankel (2004), which was published following a controversy with Seth Stein and others on the New Madrid earthquake hazard. However, our paper is by no means a contribution that that controversy. However, since the New Madrid earthquake zone may be qualified as a Slow Deforming Region, one sentence addressing the debate was included in the discussion section.

"*Taking into consideration the space geodetic information, the New Madrid seismic zone also classifies as a Slow Deforming Region (e.g. Newman et al., 1999). Inferred tectonic strain imply that the 1811, 1812, if M=8, have very low probability of occurrence (ibid.). The interpretation of this earthquake sequence generated a long-standing debate (e.g. Frankel, 2004, Stein, 2005, Frankel, 2005). The New Madrid events show that time ergodicity, a common assumption for earthquake generation in Slow Deforming Regions, can fail dramatically.*"

*Frankel, A. (2005). Reply to "Comment onHow Can Seismic Hazard in the New Madrid Seismic Zone Be Similar to That in California?'by Arthur Frankel". Seismological Research Letters, 76(3), 366-367.*

*Stein, S. (2005). Comment on "How can seismic hazard in the New Madrid Seismic Zone be similar to that in California?" by Arthur Frankel. Seismological Research Letters, 76(3), 364-365.*

---

## Author Response (AR2)

**Author comments / Letter to the editor and reply to Referee #2 comments (Robert J. Geller) Submitted on 06 Aug 2021**

Title: A sanity check for earthquake recurrence models used in PSHA of slow deforming regions: the case of SW Iberia
Author(s): Margarida Ramalho, Luis Matias, Marta Neres, Michele M. C. Carafa, Alexandra Carvalho, and Paula Teves-Costa

**Interactive comment by Robert J. Geller (Referee)**

**General comment: PSHA or no PSHA: our position**

In the revised document we made all efforts to clarify our position in relation to Robert Geller (RG) comments, namely that this paper is not a "sanity check" on PSHA but on one of its components, the earthquake recurrence models. In fact, our work can be considered in line with PSHA, and it is one suggestion for its improvement in slow deforming regions, despite the criticism that the PSHA has been subjected to. If by this assumed position in favor of an improved PSHA the paper is rejected, then we do not expect any more papers on PSHA to be published from now on and we will move to another subject. Our understanding is that the discussion is still open, and our paper could be considered a contribution to it, from the PSHA side.

Despite the author's position, we tried to acknowledge the major criticism to PSHA earthquake recurrence models and to point out its limitations. We follow some of the references that propose to abolish PSHA in endorsing the need that a wide range of users concerned with the need to build safety are involved in the decision making where the uncertainties and assumptions of the models used are clearly stated.

Winston Churchill once said that: "democracy is the worst form of government – except for all the others that have been tried."

**Reply to Referee #2 comments (Robert J. Geller) Submitted on 06 Aug 2021**

1a. The title of the paper is "A sanity check for earthquake recurrence models…." When we talk about a sanity check for a human being, we're talking about finding out whether or not the person is insane. We don't normally order a "sanity check" on a human unless we strongly suspect the person is insane. When we make a "sanity check" of PSHA this is because we strongly suspect PSHA is fundamentally flawed, i.e., that it is an incorrect theory that therefore gives incorrect results.

R: This paper is not a "sanity check" on PSHA and it will not provide a definite opinion on it. In the paper the term "sanity check" is used with the same meaning that was used, for example, for probabilistic tsunami hazard assessment in Basili et al. (2021). It means that the model predictions are tested against observations and/or independent parameter estimates, like we do in the paper. One sentence in the introduction was added clarifying this issue. The discussion on the flawness of PSHA is deferred to the new Discussion section.

1b. Two papers (Stein et al., 2012; Mulargia et al., 2017) are cited here, but not correctly. Lines 43-45 say "One of the evolutions suffered by PSHA studies and now recognized as essential, is the evaluation of the uncertainties on the results (e.g. Frankel, 2004, Stein et al., 2012, Mulargia et al., 2017), being a consequence of our incomplete knowledge of the earthquake generation and propagation mechanisms. This is a misrepresentation, for the 2017 paper in particular. The first paper (Stein et al., 2012) points out that, for whatever reason, many damaging earthquakes occurred in regions designated as being at relatively low risk by PSHA-based hazard maps. We did not say that this discrepancy is due to "our incomplete knowledge of the earthquake generation and propagation mechanisms." We just note that the discrepancy exists.

R: We acknowledge the incorrect interpretation made in the text and the introduction section was corrected.

2. The second misrepresentation is more serious. Mulargia et al. (2017) do not say the problems of PSHA are because of "our incomplete knowledge of the earthquake generation and propagation mechanisms." Rather, we say that the fundamental assumption of PSHA (that past seismicity rates can be conflated with future probabilities) is wrong, and that the use of PSHA should therefore be abandoned. The "sanity check" made in the present paper should therefore be framed as one trial to see if the position of Mulargia et al. (2017) is correct or not.

R: As mentioned above, the proposed paper cannot be considered as a trial for the position of Mulargia et al. (2017). First of all we tried to make clear in the paper that PSHA has two major sources for uncertainty (and possible causes for it to be

"wrong"), one related to the earthquake recurrence models, in space and time, and the other related to the Ground Motion Prediction Equations that translate the earthquake occurrence to ground motion. Our paper discusses only the first of these two that depends above all on the physics of earthquakes. While criticizing the lack of physical constrains in the models used for earthquake generation in slow deforming regions, we suggest procedures to improve them considering its relationship to plate kinematics, in line with the moment release balance made by Bird and Kagan (2004). This subject is discussed on the new Discussion section.

3. The authors (lines 24-43) discuss the widespread use of PSHA by society. This discussion should also mention that the widespread use of PSHA is an example of a kind of "group-think," in as much as PSHA is being used by almost everyone without having been validated by anyone.

R: We express our opinion on the difficulties that occur in the validation of PSHA studies in the new Discussion section.

4. Conclusions section: What is the conclusion: sane or insane? This should be clearly stated.

R: As mentioned above this paper is not a "sanity check" on PSHA and it will not provide a definite opinion on it. This position should be clearer by the changes made in the Introduction and the new Discussion sections.

5a. Discussion section: A brief "discussion" section should be added.

R: Done. Maybe not so brief as suggested by R. J. Geller but, required to address the main points questioned.

5b. In particular, I recommend that Klemeš (1989) (as well as, perhaps, Klemeš, 1986 as well) should be cited and discussed. His work discusses hydrology, but the problem in our field is the same. We have the PSHA machinery into which we put numbers, and like magic, other numbers are output. But this is just a mathematical pastime and the numbers that come out are just numbers, nothing more, unless the model used by PSHA has been validated. In fact, not only has it not been validated, but, as pointed out by Mulargia et al. (2017) it appears to be wrong. So what should everyone involved in risk and hazard estimation be doing instead of blind (or conditional) reliance on PSHA? Please discuss this.

R: We found the Klemeš, 1986 reference worth to be mentioned in the new Discussion section. However, our interpretation of its arguments differs from R. J. Geller ones.

As a final note, not included in the paper, we may note that Mulargia et al. (2017) provides some hints on the procedures to replace PSHA. It uses terms as "earthquake resistance" and "sufficient strength" adding to "including beyond current-code earthquakes". However, there is no indication whatsoever on the methods to define the earthquakes used to evaluate the "resistance" or "strength" of the buildings, nor the ways to choose the earthquakes "beyond the codes". Our belief, expressed in the discussion, is that even if PSHA will not be the single tool to make the required for land use management and for the establishment of building codes, it will still remain has one of the tools to be used and for this reason, any effort to improve it should be considered. Incorporating physics on a blind statistical model was our task in this paper.

---

## Author Response (AR3)

**Author's response / Letter to the editor**

**Title: A sanity check for earthquake recurrence models used in PSHA of slow deforming regions: the case of SW Iberia**
**Author(s): Margarida Ramalho, Luis Matias, Marta Neres, Michele M. C. Carafa, Alexandra Carvalho, and Paula Teves-Costa**

**Interactive comment by Robert J. Geller (Referee)**

**Our renewed acknowledgements to Referee #2 Robert J. Geller**

We address a special thanks to Robert J. Geller for helping us bringing the paper to an higher level of interest for the research community, that we initially only mentioned lightly.

**Reply to Referee #2 comments (Robert J. Geller) Submitted on 10 Nov 2021**

1. Add to line 56 after "….uncertainties." and before "While…..": Mulargia et al. (2017) point out that PSHA assumes that frequencies of past earthquakes can be conflated with probabilities of future earthquakes, but that this assumption appears to be incorrect. All researchers should be aware of this issue.

R: We add the statement as suggested.

2. Add to end of line 689: One possible hope for objectively evaluating the results of PSHA would be aggregating results globally, as was done by Rong et al. (2003) to evaluate the seismic gap model and show that its predictions were not statistically significant. In any case, policy makers and stakeholders should be aware that although PSHA is widely used, it has not yet been validated by objective testing. Thus PSHA should not be relied on as a "black box."

R: As proposed, we add the above sentence, as well as the suggested additional reference of Rong et al., 2003.